



# Development of the Community Water Model (CWatM v1.04) A high-resolution hydrological model for global and regional assessment of integrated water resources management

Peter Burek[1], Yusuke Satoh[1,2], Taher Kahil[1], Ting Tang[1], Peter Greve[1], Mikhail Smilovic[1], Luca Guillaumot[3] and Yoshihide Wada[1,4]

[1] International Institute for Applied Systems Analysis, Laxenburg, Austria
[2] National Institute for Environmental Studies, Tokyo, Japan
[3] Univ Rennes, CNRS, Géosciences Rennes - UMR 6118, F-35000 Rennes, France
[4] Department of Physical Geography, Utrecht University, Utrecht, Netherlands

Correspondence to: Peter Burek (burek@iiasa.ac.at)

**Abstract**

We develop a new large-scale hydrological and water resources model, the Community Water Model (CWatM), which can simulate hydrology both globally and regionally at different resolutions from 30 arc min to 30 arc sec at daily time steps. CWatM is open-source in the Python programming environment and has a modular structure. It uses global, freely available data in the netCDF4 file format for reading, storage, and production of data in a compact way. CWatM includes general surface and groundwater hydrological processes, but also takes into account human activities, such as water use and reservoir regulation, by calculating water demands, water use, and return flows. Reservoirs and lakes are included in the model scheme. CWatM is used in the framework of the Inter-Sectoral Impact Model Intercomparison Project (ISIMIP), which compares global model outputs. The flexible model structure allows dynamic interaction with hydro-economic and water quality models for the assessment and evaluation of water management options. Furthermore, the novelty of CWatM is its combination of state-of-the-art hydrological modeling, modular programming, an online user manual and automatic source code documentation, global and regional assessments at different spatial resolutions, and a potential community to add to, change, and expand the open-source project. CWatM also strives to build a community learning environment which is able to freely use an open-source hydrological model and flexible coupling possibilities to other sectoral models, such as energy and agriculture.

## 1 Introduction

In recent years, the interactions between natural water systems, climate change, socioeconomic impacts, human management of water resources, and ecosystem management have increasingly been incorporated into the processes of large-scale hydrological models (Wada et al., 2017). Examples of these models are WaterGAP (Alcamo et al., 2003;Flörke et al., 2013), H08 (Hanasaki et al., 2008, 2018), MATSIRO (Pokhrel et al., 2012), LISFLOOD (De Roo et al., 2000;Udias et al., 2016), PCR-GLOBWB (Van Beek et al., 2011;Wada et al., 2014, Sutanudjaja et al., 2018), }, SAFRAN-ISBA-MODCOU (Habets et



al., 2008; Decharme et al., 2019). Human intervention in hydrology and water resources is becoming essential for the realistic simulation of global and regional hydrological processes. In particular, simulations of human water demands from different sectors such as agriculture, industry, and households could have a large impact on estimated hydrological storage (e.g., groundwater) and fluxes (e.g., discharge) (Alcamo et al., 2007; Wada et al., 2016). More efforts have gone into better

groundwater representation in large-scale hydrological models to realistically simulate groundwater levels and surface–groundwater interactions (Pokhrel et al., 2015;Wada, 2016; Reinecke et al., 2019; de Graaf et al., 2015, 2017; Decharme et al., 2019).

In recent years, model intercomparison projects such as the WaterMIP (Water and Global Change Water Model Intercomparison Project (Haddeland et al., 2011), Inter-Sectoral Impact Model Intercomparison Project (ISIMIP) (Warszawski

et al., 2014), and the Coupled Model Intercomparison Project Phase 6 (CMIP6) (Eyring et al., 2016) led to, among other advantages, a systematic overview of models, a consistent database of spatial input data and simulation protocol and scenarios, and a shared database of results, all of which facilitate analysis across different modeling sectors (e.g., water, agriculture, energy, biome, and climate). This has also led to a better understanding of how to assess future changes in land use and climate in relation to water resource constraints under given uncertainties of the forcing drivers such as climate.

Clark et al. (2011) and Bierkens (2015) indicate that model intercomparison efforts have failed to lead to a better understanding of the origins and consequences of systematic model bias and differences, and thus to an improved outcome of model components. Bierkens (2015) argues that while there are many catchment hydrological models for specific catchments specializing into their own sophisticated model parameterizations, few global hydrological models — compared with the number of regional hydrological models — interact with these regional models and modeling groups (e.g. Siderius et al., 2018).

One way of overcoming this barrier could be to implement multiple modeling or module approaches into the unifying framework suggested by Clark et al. (2015). Thus, we here develop a new large-scale hydrological and water resources model, the Community Water Model (CWatM), which has a flexible modular structure and unique global and regional spatial representations. Because of complex interactions of hydrology with food, energy and ecosystems, it is expected that hydrological models can cover these interactions as model components. To advance the move from large-scale hydrological

models to better model representations of hydrological processes, we believe that it is also necessary to create a community-driven modeling environment that facilitates the exchange of ideas, components or modules, data, and results in easily communicable format. In a wider sense, a user-friendly and flexible model structure will enable more active engagement with stakeholders and associated capacity training.

Therefore, CWatM includes the features detailed below:

• Use of an open-source platform as a way to exchange ideas and develop model codes that facilitate capacity enhancement, especially in regions with limited access to high computation facilities and high-resolution data

    • Scalability to allow use of the model at the regional to catchment scale and also at the continental to global scale, which facilitates learning between global and regional hydrological model applications




- Use of flexible modular structure to explore the linkages with other sectoral models such as those relating to land use,
 agriculture, and energy so that options and solution space could be integrated

- Existing linkages to state-of-the-art models for energy (MESSAGE) (Sullivan et al., 2013), land use and ecosystems
 (GLOBIOM) (Havlík et al., 2013) , agriculture (IIASA-EPIC) (Balkovič et al., 2014), water quality (MARINA) (Strokal
 et al., 2016) and hydro-economy (ECHO) (Kahil et al., 2018)

- Linkages to the political economy and stakeholder perspectives (Tramberend et al., 2019) for example, social hydrology
 (Sivapalan et al., 2012; Seidl and Barthel, 2017)

A model software architecture includes the aspects below:

- A high-level programming language for easy comprehension of the code and to facilitate extensibility

- An interface to a fast computing language (e.g., C++) for time-intensive operations (e.g., river routing)

- A multi-platform to adjust the model to the users' needs and capacity (e.g., Windows, Linux, Mac and high-performance
 clusters and super-computers).

- A high level of modularity to be extensible for different model options to solve the same process, for example, evaporation
 with Hargreaves, Hamon, Penman-Monteith or for a different purpose (e.g., flood forecasting, water–food nexus, linking
 to hydro-economic modeling).

- Documentation of the model and the source code in a semi-automatic way to facilitate immediate documentation and
 comprehension of the concepts involved

- A state-of-the-art data structure for reading and writing time/spatial data to allow efficient management of data storage
 and facilitate the development toward high resolution models

As described above, the main novelty of CWatM lies not in providing entirely new concepts for modeling hydrological and
socioeconomic processes but in combining existing good practice in various scientific communities beyond hydrology itself.
CWatM has a modular model structure which is open-source and uses state-of-the-art data storage protocols as input and output
data. Currently, CWatM can use different spatial resolution from 30 arc min (≈ 50 km by 50 km at the equator) to 30 arc sec
(≈ 1 km by 1 km) enabling it to address both global and regional water management. The online user manual and automatic
source code documentation make CWatM an easy-to-use tool which can be integrated and coupled to other toolsets such as
land use modeling and hydro-economic modeling. CWatM also strives to build up a community which can freely use an open
source hydrological model with the possibilities of coupling it to other water management models such as WEAP (Yates et al.,
2005) and ECHO (Kahil et al., 2018).

This paper describes the development of the model, including its structure and modules, and gives some examples of
applications. Section 2 of this paper presents a detailed description of the model development of CWatM. Section 3 describes
the data used for the model. Section 4 introduces the calibration of the model. Section 5 shows results for several calibrated
catchments and two application examples. Section 6 discusses the conclusions and the way forward.





## 2 Model description

### 2.1 Model concept

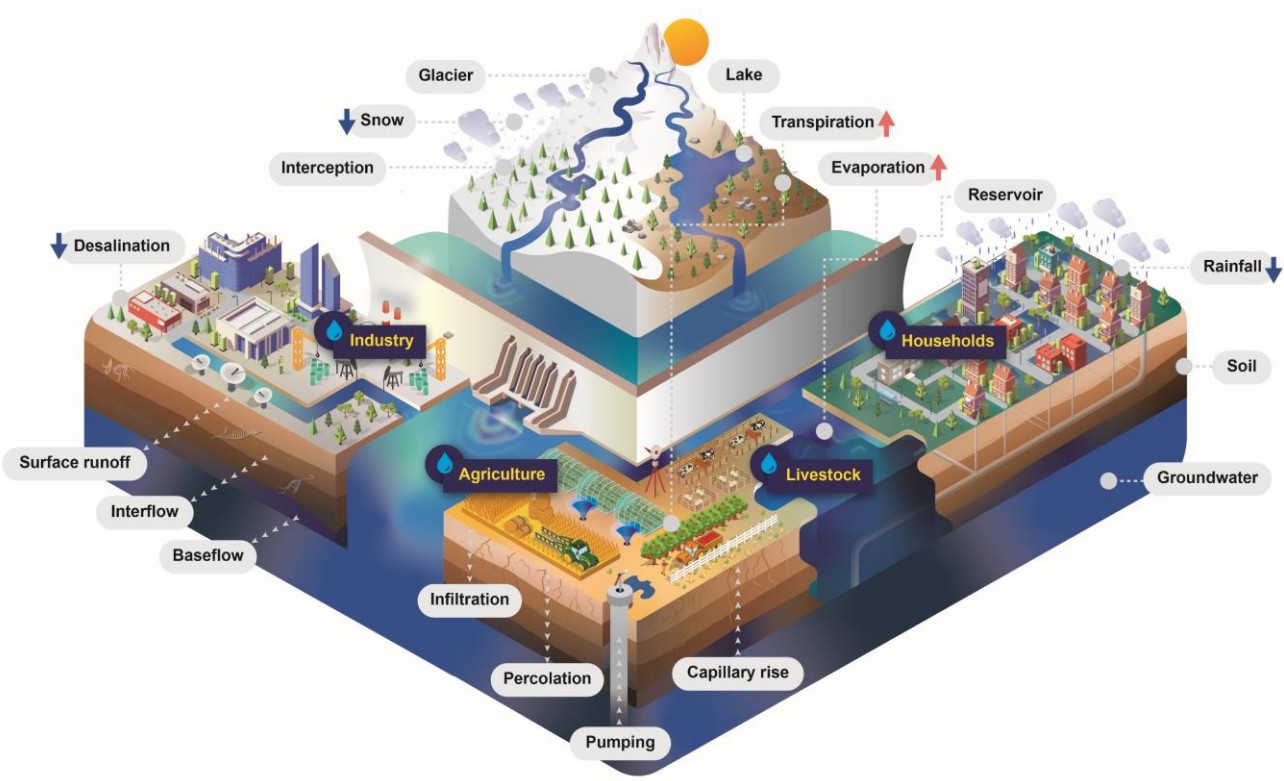

**Figure 1: Schematic figure of the processes included in the CWatM**

The Community Water Model (CWatM) is an integrated hydrological and channel routing model developed at the International Institute for Applied Systems Analysis (IIASA). CWatM quantifies water availability, human water use, and the effect of water infrastructure, for example, reservoirs, groundwater pumping, and irrigation, in regional water resources management. A schematic view of the processes is given in Figure 1. CWatM is a grid-based model with recent version for spatial resolution

of 0.5° and 5' (with sub-grid resolution taking into account topography and land cover) at daily resolution (with sub-daily time stepping for soil, lakes and reservoirs and river routing). The model can also be applied at 30 arc sec. CWatM follows a modeling concept similar to that of large-scale hydrological models such as H08 (Hanasaki et al., 2006;2008; 2018), WaterGAP (Alcamo et al., 2003;Flörke et al., 2013), LPJmL (Bondeau et al., 2007; Rost et al., 2008), LISFLOOD (De Roo et al., 2000;Burek et al., 2013;De Roo et al., 2000), PCR-GLOBWB (Van Beek et al., 2011;Wada et al., 2014;Sutanudjaja et al.,

2018), VIC (Xu et al., 1994), MHM (Samaniego et al., 2011;Kumar et al., 2013), and HBV (Bergström and Forsman, 1973;Lindström, 1997). A comprehensive overview of existing GHMs is given in Bierkens (2015), Kauffeldt et al. (2016), Pokhrel et al. (2016), Wada et al. (2017), and in the ISI-MIP project (Frieler et al., 2016) , the latter having been used for



model comparison of different GHMs. Among these large-scale hydrological models, CWatM uses a model implementation similar to that of PCR-GLOBWB and LISFLOOD.


The philosophy of CWatM is the same as that described in Bergstrom (1991) for the model HBV: as complex as necessary but not more. This means that the model merges conceptual and physical modeling and is keeping a similar level of physical complexity throughout the model. If a higher detail of physical model is needed it should be introduced as add-on modules. For different tasks, different interactions to other models and different descriptions of processes are needed.

The CWatM modeling system is written in Python 3.7 with only a few Python packages (numpy, scipy, gdal, netCDF4) and can be used on different platform (Unix, Linux, Window, Mac). Excessive computational parts can be added via an interface as C++ or Fortran code. For example, runoff concentration within a grid cell or river routing using the kinematic wave equation is done in C++. With this approach the advantage of high-level languages like Python to write and understand code fast and effective and the advantage of languages like C++ for fast computing are preserved. In order to reach a broader community

beyond hydrology, the entire model development is open source under a General Public License (GNU) with a repository on GitHub (https://github.com/CWatM/CWatM) and an online documentation including documentation on the source code on IIASA webpage (https://cwatm.iiasa.ac.at) featuring github.io.

The focus of the model development is to build a flexible model architecture and to present a full hydrological model for calculating water availability and demand. The modular structure of CWatM is a basic principle of the entire model

development, as, for example, in separating data management (e.g., reading configuration, data read and write routines, error handling) from the hydrological modules. The hydrological modules deal only with the processes, while data input and output are performed in separate modules. The model can handle different spatial resolution from 1 km to 50 km at a daily temporal resolution for different tasks from global to regional assessments. CWatM has a modular build of separate modules for each hydrological process group (from calculation of potential evaporation to river routing). In addition, each module is identically

composed of an initialization class and a dynamic class. This idea is taken from PC-Raster framework (Karssenberg et al., 2010). The model can be run without extensive experience in Python by using a setting file, which include all relevant information of data and parameters for each process. CWatM uses netCDF4 format as in- and output to store temporal-spatial data efficiently and to use meteorological forcing data as they come without reformatting. Together with the auto-documentation of source code available in the online manual, this simplifies the understanding, modification, or exchanging

of components. This allows the model to be tailored to the needs of the user so that different research questions for different spatial units from global to local scales can be answered, something which is increasingly needed for different stakeholders and science communities beyond hydrology to enable them to engage with each other.





## 2.2 General overview of the hydrological processes

CWatM can use different datasets of daily meteorological forcing as inputs to calculate potential evaporation with Penman-Monteith (Allen et al., 1998) as a default option, as well as other methods such as the Hargreaves (Hargreaves and Samani, 1958) and Hamon (Hamon, 1963) approaches. Elevation data on the sub-grid level and temperature are used to split precipitation into rain and snow, while the degree-day factor method (WMO, 1986) calculates snow melt.

CWatM calculates the water balance for six land cover classes separately (forest, irrigated, paddy-irrigated, water covered, sealed area and "other" land cover class). Soil processes, interception of water, and evaporation of intercepted water are calculated separately for four different land cover classes (forest, irrigated, paddy-irrigated and "other" ) and the resulting flux and storage per grid-cell is aggregated by the fraction of each land cover class in each grid-cell. Infiltration into the soil is calculated with the Xinanjiang model approach (Zhao and Liu, 1995;Todini, 1996). The model calculates preferential bypass flow which bypasses the soil layers and percolates directly to groundwater, similar to the approach of LISFLOOD (Burek et al., 2013), VarKast (Hartmann et al., 2015) and HBV (Lindström et al., 1997). Soil moisture redistribution in three soil layers is calculated using the Van Genuchten simplification (Van Genuchten, 1980) of the Richards equation. The depth of the first soil layer is fixed at 5 cm, so that its soil moisture can be compared with products from remote sensing data. The second and third soil layer depths depend on the root zone depth of each land cover class and the total soil depth from data of the Harmonized World Soil Database 1.2 (HWSD) (FAO et al., 2012). Water uptake and transpiration by vegetation are based on an approach by Supit et al. (1994) and Supit and van der Goot (2003) where water stress reduces the maximal transpiration rate. Direct evaporation from the soil surface is calculated separately . For two more land cover classes, namely, water and sealed (impermeable) surface, evaporation and runoff are also calculated separately.

Groundwater storage is modeled using a linear reservoir. In the newest version of the model, a MODFLOW coupling is also available, allowing to include lateral flows between grid cells. Capillary rise from groundwater to the soil layers is included. Runoff concentration in a grid-cell is calculated using a triangular-weighting-function. CWatM applies the kinematic wave approximation of the Saint-Venant equation (Chow et al., 1998) for river routing.

Lakes and reservoirs are included in two different ways: i) a lake or reservoir has an upstream area beyond the actual grid cell and is part of the grid linking the river routing system; ii) a lake or reservoir is only a part of the regional river system within a grid cell. Reservoirs are simulated using a simple general reservoir operation scheme as used in LISFLOOD (De Roo et al., 2000, Burek et al., 2013). Lakes are simulated by using the Modified Puls approach (Chow et al., 1998, Maniak, 1997).

Water demand and consumptions are estimated for the livestock, industry, and domestic sectors using the approach of Wada et al. (2011,2014). Water demand and consumption for irrigation and paddy irrigation are calculated within CWatM using the crop water requirement methods of Allen et al. (1998). This irrigation scheme can also dynamically link the daily surface and soil water balance with irrigation water.





With these coupled processes CWatM can facilitate assessment of the changing pattern of water supply and demand across scales under climate change at different spatial resolutions. The modular structure also makes possible the linking of CWatM with other IIASA models, for example, MESSAGE (Sullivan et al., 2013), GLOBIOM (Havlík et al., 2013), and ECHO (Kahil et al., 2018) to develop an integrated assessment modeling framework for nexus issues (e.g., water–food–energy) or hydro-economic modeling for quantifying water infrastructure investment options for regional water resources management.

## 2.3 Methods

### 2.3.1 Meteorological forcing

CWatM is able to use different dataset of meteorological forcing for current climate, for example, MSWEP (Beck et al., 2017), WFDEI (Weedon et al., 2014), PGMFD (Sheffield et al., 2006), GSWP3 (Kim et al., 2012) or EWEMBI (Lange, 2018) or future climate projections from different General Circulation Models (GCMs), ) (e.g., data from ISIMIP project (Frieler et al., 2016). CWatM can use the netCDF4 repositories of original meteorological forcing without reformatting.

Depending on the method used for calculating potential evaporation e.g., Penman-Monteith method (Allen et al., 1998), Hargreaves method (Hargreaves and Samani, 1958) or Hamon method (Hamon, 1963) different climate data are needed. As a default, the Penman-Monteith needs as inputs precipitation, humidity, long- and short-wave downward surface radiation fluxes, maximum, minimum, and average 2m temperature, 10-m wind speed, and surface pressure. Temperature data are additionally needed to distinguish between snow and rain.

### 2.3.2 Potential evaporation

. Potential reference crop evaporation rate ($ET0$) is calculated from a hypothetical reference vegetation with specific characteristics and unlimited availability of water (Allen et al., 1998). In the same way the potential evaporation of an open water surface ($EW0$) is calculated. $ET0$ and $EW0$ are treated as pure climatic variables, because for calculation purpose they are not influenced by land cover or soil properties. In reality, the potential evapotranspiration might be different to $ET0$ due to differences in vegetation characteristics, aerodynamic resistance, and surface reflectivity (albedo). To account for the variariability of vegetation, the $ET0$ is multiplied by an empirical "crop coefficient" ($k_{crop}$), that lumps these differences into one factor, yielding a potential crop evapotranspiration rate ($ET_{crop}$). The method used here is based on work described in Allen et al. (1998) and Supit and van der Goot (2003).

### 2.3.3 Snow and frost

Precipitation is split into rainfall and snow, depending on the temperature. If the average temperature is below 1°C (default, but can be changed), all precipitation is assumed to be snow. For large grid cells, of, say, 0.5° or 5' resolution, there is a





considerable sub-grid heterogeneity in elevation and therefore in temperature and snow accumulation and melt (Anderson, 2006). Because of this, snow accumulation and melt are modeled in up to 10 separated elevation zones on sub-grid level using

different elevation zones and a fixed defined moist adiabatic lapse rate.

Snow accumulates until it melts or evaporates. The rate of snowmelt is estimated using a degree-day factor method which take into account that snowmelt increases when it is raining (Speers and Versteeg, 1979)

$$M = C_m \cdot C_{Seasonal}(1 + 0.01 \cdot R\Delta t)(T_{avg} - T_m) \cdot \Delta t \tag{1}$$

where:
M:        snowmelt per time step [mm]
      R:        rainfall intensity [mm day$^{-1}$]
      $\Delta$t:       time interval [days]
      $T_m$:       = 0 °C
      $C_m$:       degree-day factor [mm °C$^{-1}$ day$^{-1}$]
$C_{Seasonal}$:  seasonal variable melt factor

$C_{Seasonal}$ is a factor depending on the day of the year, which varies the snow melt rate. A similar factor is used in several other models (e.g., Anderson, 2006 and Viviroli et al., 2009). At high altitudes the model tends to accululate snow without any melting loss, because temperature never exceeds 1°C,. In these altitudes snow is accumulated and is converted into firn, which then is converted to ice and as glacier moved to lower regions over decades or even centuries.. In the ablation area, the ice is

again melted. In CWatM this process can be optionally simulated by melting the snow in higher altitudes on an annual basis over summer using a higher degree-day factor and temperature from a lower sub-grid zone.

Hydrological processes occurring near the soil surface are affected and halted, if the soil surface is frozen. To estimate whether the soil surface is frozen, A frost index $F$ is calculated to estimate whether the soil surface is frozen based on Molnau and Bissell (1983). The frost index changes by day at a rate given by:

$$\frac{dF}{dt} = -(1 - A_f)F - T_{av} \cdot e^{-0.04 \cdot K \cdot d_s/we_s} \tag{2}$$

where:
      $dF/dt$:   [°C day$^{-1}$].
      $A_f$:      decay coefficient [day$^{-1}$] (here: 0.97)
      $K$:       snow depth reduction coefficient [cm$^{-1}$] (here 0.57)
$d_s$:      grid-average of depth of the snow cover [mm equivalent water depth]
      $we_s$:     snow water equivalent

For each time step the value of $F$ [°C day$^{-1}$] is updated as:

$$F(t) = F(t - 1) + \frac{dF}{dt}\Delta t \tag{3}$$

The soil is considered frozen when the frost index is above a critical threshold of 56 and every soil process in the first two

layers will be stopped. Precipitation bypasses soil and is transformed into surface runoff until the frost index is again lower than 56.



### 2.3.4 Interception, evaporation from soil, open water, and sealed surface

The calculation for interception and evaporation is based on Allen et al. (1998). For each land cover class, a maximum interception storage is defined. Interception storage can be filled by rainfall and depleted by evaporation using potential

evaporation from open water. The left-over interception storage is added to the water available for infiltration in the other time step. Evaporation from soil is calculated using the potential reference evapotranspiration rate multiplied by a soil crop factor (default: 0.2). Evaporation from sealed area or open water is calculated using the potential evapotranspiration for the open water rate multiplied by a factor (default: 0.2 sealed, 1.0 water).

### 2.3.5 Transpiration from plants

Potential transpiration from plants is calculated using the potential reference evapotranspiration multiplied by a crop-specific factor available as a spatially distributed data set for each land cover type for every 10 days over a year. The crop coefficient is aggregated from MIRCA2000: a global data set of monthly irrigated and rainfed crop areas (Portmann et al., 2010). The actual transpiration rate depends on the available water and on the ability of the crop type to deal with water stress. The energy already used up for the evaporation of intercepted water is subtracted here in order to respect the overall energy balance. The

actual transpiration rate is reduced by a water stress factor which takes into account the ability of the crop to deal with water stress and an index of water stress of the soil:

$$r_{WS} = \frac{(w_1 - w_{wp1})}{(w_{crit1} - w_{wp1})} \tag{4}$$

where:

$r_{ws}$:      Reduction factor because of water stress
$w_1$:      soil moisture in the two upper soil layers [mm]
$w_{wp1}$:      soil moisture at wilting point (soil moisture potential pF 4.2)
$w_{crit1}$:      soil moisture below which water uptake is reduced and plants start closing their stomata

The critical amount of soil moisture is calculated as:

$$w_{crit1} = (1 - p) \cdot (w_{fc1} - w_{wp1}) + w_{wp1} \tag{5}$$

$\quad$ p=1 /(0.76+1.5 \cdot T_{max}) - 0.1 \cdot (5 \cdot Crop_{Groupnumber})$

where:

p:      soil depletion fraction
$w_{fc1}$:      soil moisture at field capacity
Crop_{Groupnumber}:      The crop group number is an indicator of adaptation to dry climate (e.g., olive groves are adapted

to dry climate and can therefore extract more water from soil that is drying out than rice can.

The actual transpiration $T_a$ is calculated:

$$T_{act} = r_{WS} \cdot T_{pot} \tag{6}$$

The procedure for estimating *p and $R_{ws}$* is described in detail in Supit and van der Goot (2003).





### 2.3.6 Infiltration into soil and preferential bypass flow

To estimate the infiltration capacity of the soil the approach of XinanJiang (also known as VIC/ARNO model) (Zhao and Liu, 1995 and Todini, 1996) is used. The saturated fraction of a grid cell that contributes to surface runoff is related to the overall soil moisture of a grid cell through a non-linear distribution function. The saturated fraction $A_s$ is approximated by the following distribution function:

$$A_s = 1 - (1 - \frac{w_1}{w_{s1}})^b \qquad (7)$$

where:
$w_{s1}$, $w_1$: maximum and actual soil moisture in the upper two soil layer
b: empirical shape parameter

$$INF_{pot} = \frac{w_{s1}}{b+1} - \frac{w_{s1}}{b+1}[1 - (1 - A_s)^{\frac{b+1}{b}}] \qquad (8)$$

To simulate the preferential bypass flow of the soil, a fraction of the water available for infiltration is passed directly to the groundwater zone. The fraction is calculated as a function of the relative saturation of the first two soil layers.

$$D_{pref,gw} = W_{av}(\frac{w_1}{w_{s1}})^{c_{pref}} \qquad (9)$$

where:
$D_{pref,gw}$: preferential flow per time step
$W_{av}$: available water for infiltration
$c_{pref}$: empirical shape parameter

A preferential flow component is calculated, that lets more water bypass the soil as the soil gets wetter.

The actual infiltration $INF_{act}$ is calculated as:

$$INF_{act} = min(INF_{pot}, W_{av} - D_{pref,gw}) \qquad (10)$$

### 2.3.7 Soil moisture redistribution and capillary rise

Unsaturated flow and transport processes are modeled with the 1D-Richard equation, which requires a high spatial and temporal distribution of the soil's hydraulic properties.

$$\frac{\Delta\theta}{\Delta t} = \frac{\Delta}{\Delta z}\left[K(\theta)\left(\frac{\Delta h(\theta)}{\Delta z}\right) - 1\right] - S(\theta) \qquad \text{(1D Richard equation)} \qquad (11)$$

where:
θ: soil volumetric moisture content [L3/L3]
t: time [T]
h: soil water pressure head [L]
K(θ): unsaturated hydraulic conductivity [L/T]
z: vertical coordinate
S: source sink term [T-1]





With a simplification, the 1D Richard equation (e.g., flow of soil moisture), is entirely gravity-driven and the matrix potential gradient is zero. This implies a flow that is always in a downward direction at a rate equal to the conductivity of the soil. The relationship can now be described with the model of (Mualem (1976) and with the Van Genuchten model equation.

$$K(\theta) = K_S \left(\frac{\theta - \theta_r}{\theta_s - \theta_r}\right)^{0.5} \left\{1 - \left[1 - \left(\frac{\theta - \theta_r}{\theta_s - \theta_r}\right)^{1/m}\right]^m\right\}^2 \qquad \text{(Van Genuchten equation)} \qquad (12)$$

where:
Ks:        saturated conductivity of the soil [cm/d-1]
$K(\theta)$:        unsaturated conductivity
$\Theta$, $\theta_s$, $\theta_r$: actual, maximum and residual amounts of moisture in the soil [mm]
m:        calculated from the pore-size index ($\lambda$): $m = \frac{\lambda}{\lambda + 1}$


The soil hydraulic parameters $\Theta$, $\theta_s$, $\theta_r$, $\lambda$, and $K_s$ are needed to simulate soil water transport for the Van Genuchten model and are derived via a pedotransfer function, (e.g., model Rosetta of: Zhang and Schaap, 2017) from standard soil properties (soil texture, porosity, organic matter and bulk density).

Once the unsaturated conductivity for each soil zone is determined, the water flux to the next zone can be estimated. At a time
step of one day and high $K(\theta)$, the vertical flux can exceed the available soil moisture:

$$K(\theta) > \theta - \theta_r \qquad (13)$$

Therefore, the soil moisture equation has to be solved iteratively on a sub-daily time step.

Capillary rise occurs only when the groundwater level is close to the surface. CWatM estimates the total fraction of the area with groundwater level of between 0m and 5m from the surface in discrete steps and calculates the flux from groundwater to
the soil layer based on unsaturated conductivity and field capacity (Wada et al., 2014).

### 2.3.8 Groundwater

Groundwater storage and baseflow are modeled using a linear reservoir approach as in LISFLOOD (De Roo et al., 2000;Udias et al., 2016). The groundwater zone is filled by the water percolating from the lower soil zone and the preferential flow and is emptied by capillary rise and baseflow. The outflow from the groundwater zone is given by:

$$Q_{base} = \frac{1}{T_{base}} Storage = R_{coeff} Storage \qquad (14)$$

where:
$Q_{base}$:        Baseflow or outflow from the groundwater zone
$T_{base}$:        Groundwater reservoir constant in days
Storage: Water stored in the groundwater zone
$R_{coeff}$:        Recession coefficient of groundwater zone

For considering lateral fluxes among grid cells and the explicit computation of groundwater levels over finer spatial domains, CWatM has an option to couple with MODFLOW (McDonald and Harbaugh, 1988, Harbaugh, 2005) using the FloPy Python package (Bakker et al., 2016) in a similar way to the PCR-GLOBWB (Sutanudjaja et al., 2014). The 5' resolution version of





CWatM is coupled with an one-layer MODFLOW model at a finer MODFLOW resolution (from 4 km to 400 m) with the aim
of integrating the small-scale topographic control. The coupling is made on a daily to weekly base water balance.

CWatM simulates the vertical soil water flow in three soil layers, while MODFLOW simulates lateral groundwater flows. The
CWatM-MODFLOW is technically coupled (using the Drain package) via capillary rise from groundwater to the soil zones,
groundwater recharge from the soil zones, and baseflow outflow from groundwater to the river network system. As
MODFLOW resolution can be smaller than CWatM resolution, CWatM mesh is subdivided into two parts: one part where
groundwater recharge occurs and one part where capillary rise from groundwater occurs. The area of each part is determined
by the percentage of MODFLOW cells, where the water level reaches the lower soil layer inside a CWatM mesh. To distinguish
whether the groundwater flow to the surface will be attributed to capillary rise or baseflow, a percentage of rivers is attributed
to each MODFLOW cells and calculated based on a 200m resolution topographic map. Aquifer properties, like transmissivity
or aquifer thickness, are estimated using the approach of de Graaf et al. (2015) and Gleeson et al. (2011). The results presented
in section 5 of this work are calculated using the simplified linear reservoir approach.

### 2.3.9 Runoff concentration within a grid-cell

The process between runoff generation and river routing for each grid cell is called runoff concentration. The runoff generated
from each cell is routed to the corner of each cell. Depending on land cover class, slope, and runoff group (surface, interflow,
or baseflow) a concentration time (peak time) is determined. The total runoff for a grid cell is then calculated using a triangular-
weighting-function:

$$Q(t) = \sum_{landcover} \sum_{runoff} \sum_i^{max} c(i)\, Q_{runoff}(t - i + 1) \tag{15}$$

where:

| | |
|---|---|
| Q(t): | total runoff of a grid cell of a timestep |
| runoff: | runoff component (surface, interflow, baseflow) |
| Qrunoff: | runoff of land cover class of a runoff component |
| t: | time (1 day) |

c(i):        Triangular function: $c(i) = \int_{i-1}^{i} \frac{2}{max} - \left| u - \frac{max}{2} \right| \cdot \frac{4}{max^2}\ du$    (16)

### 2.3.10 River routing

Flow through the river network is simulated using kinematic wave equations. The basic equations used are the equations of
continuity and momentum. The continuity equation is:

$$\frac{\Delta Q}{\Delta x} + \frac{\Delta A}{\Delta t} = q \tag{17}$$

where:
Q: channel discharge [m³ s-1],
A: cross-sectional area of the flow [m2]
q: amount of lateral inflow per unit flow length [m2 s-1]





The momentum equation can also be expressed as in Chow et al. (1998):

$\quad A = \propto \cdot Q^{\beta}$ (18)

The coefficients α and β are calculated by putting in Manning's equation. This leads to a nonlinear implicit finite-difference solution of the kinematic wave if you transform the right side:

$$\frac{\Delta t}{\Delta x} Q_{i+1}^{j+1} + \propto \left(Q_{i+1}^{j+1}\right)^{\beta} = \frac{\Delta t}{\Delta x} Q_i^{j+1} + \propto \left(Q_{i+1}^{j}\right)^{\beta} + \Delta t \left(\frac{q_{i+1}^{j+1} + q_{i+1}^{j}}{2}\right)$$ (19)

where:
$\quad$ J: $\quad$ time index
$\quad$ I: $\quad$ space index
$\quad$ α, β: $\quad$ coefficients

With the coefficient α, β coefficient, the non-linear equation can be solved for each grid cell and for each time step using an iterative approach given in Chow et al. (1998). The coefficients can be calculated using Manning's equation.

$\quad A = \left(\frac{n \cdot P^{2/3}}{\sqrt{S_o}}\right)^{3/5} Q^{3/5}$ (20)

where:
$\quad$ n: $\quad$ Manning's roughness coefficient
$\quad$ P: $\quad$ wetted perimeter of a cross-section of the surface flow [m]
$\quad$ $S_0$: $\quad$ topographical gradient

Solving this for α and β gives:

$\quad \propto = \left(\frac{n P^{2/3}}{\sqrt{S_o}}\right)^{\beta} \ and \ \beta = 0.6$ (21)

where:
$\quad$ P: $\quad$ wetted perimeter approximated in CWatM: P = channel width + 2 * channel bankful depth
$\quad$ n: $\quad$ Manning's coefficient
$\quad$ $S_0$: $\quad$ gradient (slope) of the water surface: S0 = Δ elevation/channel length

To calculate α, CWatM uses a fixed network depending on the spatial resolution and, for each grid cell, the channel width, depth, length, gradient, and Manning's roughness have to be known.

**2.3.11 Reservoirs and lakes**

$\quad$ Reservoirs and lakes (RL) based on the HydroLakes database (Messager et al., 2016;Lehner et al., 2011) are simulated as part of the channel network. Using the approach of Hanasaki et al. (2018) and Wisser et al. (2010) we distinguish between global RL and local RL. Global RL are located in the main channel of a grid cell with a catchment upstream of this grid cell. Local RL are more or less situated inside one grid cell at the tributaries of the main channel and not attached to the main river. Local RL are defined in CWatM depending on the spatial resolution. All RL with an RL area of less than 200 km² at 0.5° (5 km² for





5') or with a watershed of less than 5000 km² at 0.5° (200 km² for 5') are defined as "global" RL. The approach to calculating water storage and outflow of RL are the same for local and global RL, but the retention effect of local RL will be calculated during the runoff concentration process within a grid cell, while the effect of global RL will be calculated during the river routing process and includes the whole river network of a catchment.

**Reservoir operation method**

The method of simulating reservoir operations is taken from LISFLOOD (Burek et al., 2013). A total storage capacity $S$ is assigned to each reservoir, and the fraction of filling of a reservoir is calculated. Three filling levels are defined: (a) the "conservative storage limit" fraction because a reservoir should never be completely empty (default set to 10% of the total storage). For prevention of damages in case of flooding a reservoir should not be filled to the full storage capacity; (b) the "flood storage limit" ($L_f$) represents this maximum-allowed storage fraction (default set to 90% of the total storage); (c) the

"normal storage limit" ($L_n$) defines the buffering capacity and the available storage of a reservoir between $L_n$ and $L_f$.

Another three parameters define how the outflow of a reservoir is regulated. (a) Each reservoir has a "minimum outflow" $O_{min}$. The default is set to 20% of the average discharge, for example, for ecological reasons. (b) A maximum possible outflow or the "non-damaging outflow" $O_{nd}$ is defined which causes no problems downstream in the case of flood. The default for this outflow is set to 400% of the average discharge. (c) Between the state of flood and normal storage limit, a reservoir is managed

as much as possible to deliver a constant outflow so that there is also a constant energy output from hydropower generation. "Normal outflow'" $O_{norm}$ is set as a default value to average discharge.

The outflow $O_{res}$, is calculated depending on the fraction of the filling of the reservoir as:

$$O_{res} = min(O_{min}, \frac{1}{\Delta t} F \cdot S) \qquad\qquad F \leq 2L_c \qquad\qquad (22)$$

$$O_{res} = O_{min} + (O_{norm} - O_{min}) \left(\frac{F - 2L_c}{L_n - 2L_c}\right) \qquad\qquad L_n \geq F > 2L_c \qquad\qquad (23)$$

$$O_{res} = O_{norm} + \frac{(F - L_n)}{(L_f - L_n)} \cdot max[(I_{res} - O_{norm}), (O_{nd} - O_{norm})] \qquad\qquad L_f \geq F > L_n \qquad\qquad (24)$$

$$O_{res} = max(\frac{(F - L_f)}{\Delta t} S, O_{nd}) \qquad\qquad F > L_f \qquad\qquad (25)$$

where:
S:        Reservoir storage capacity [m³]
F:        Reservoir fill (fraction, 1 at total storage capacity) [-]
$L_c$:        Conservative storage limit [-]
$L_n$:        Normal storage limit [-]
$L_f$:        Flood storage limit [-]
$O_{min}$:        Minimum outflow [m³ s⁻¹]
$O_{norm}$:        Normal outflow [m³ s⁻¹]
$O_{nd}$:        Non-damaging outflow [m³ s⁻¹]
$I_{res}$:        Reservoir inflow [m³ s⁻¹]





**Lake method**

Lakes are simulated using the Modified Puls approach (Chow et al., 1998, Maniak, 1997) similar to the approach as in LISFLOOD (Burek et al., 2013). As lake inflow the channel flow upstream of the lake location is used. As lake evaporation

the potential evaporation rate of an open water surface is taken. The Modified Puls approach assumes that lake retention is a special case of flood retention with horizontal water level and the equations of river channel routing (see section 2.3.10 river routing) can be written as:

$$\frac{(Q_{In1}+Q_{In2})}{2} - \frac{(Q_{Out1}+Q_{Out2})}{2} = \frac{(S_2+S_1)}{\Delta t} \tag{26}$$

where:

$Q_{In1}$:    Inflow to lake at time 1 (t)
$Q_{In2}$:    Inflow to lake at time 2 (t+Δt)
$Q_{Out2}$:    Outflow from lake at time 1 (t)
$Q_{In2}$:    Outflow from lake at time 2 (t+Δt)
$S_1$:    Lake storage at time 1 (t)
$S_2$:    Lake Storage at time 2 (t+Δt)

The change in storage is inflow minus outflow and open water evaporation. The equation is solved by calculating the lake storage curve as a function of sea level S = f(h) and the rating curve as a function of sea level Q=f(h). Lake storage and discharge are linked by the water level.

The assumptions made here to simplify the equation are:

1.) A modification of the weir equation of Poleni from Bollrich and Preißler (1992):
$$Q = \mu c b \sqrt{2g} \cdot H^{3/2} = \alpha \cdot H^{3/2} \tag{27}$$

2.) If the weir does not have a rectangular form but a parabola form, the equation can be simplified to:
$$Q = \alpha \cdot H^2 \tag{28}$$

3.) The lake storage function is simplified to a linear relation:
$S = A \cdot H$ where: S: lake storage; A: lake area; H: sea level $\tag{29}$

**2.3.12 Water use module**

**Irrigation water demand**

Irrigation is by far the biggest consumer of water at around 70% of global gross water demand (Döll et al., 2009). Irrigation

water demand is calculated following the method developed in PCR-GLOBWB (Wada et al., 2011, 2014) using the crop calendar of Portmann et al. (2010) and irrigated areas from Siebert et al. (2005) to account for seasonal variability, different crops and different climatic conditions. We refer to Wada et al. (2014) for the detailed descriptions. In brief, irrigation water withdrawal and consumption are calculated separately for paddy (rice) irrigation and irrigation of other crops. To represent flooding irrigation of paddy fields, a 50 mm surface water depth is maintained until a few weeks before the harvest. Paddy

irrigation demand is a function of the storage change of the surface water layer, net precipitation, infiltration to lower soil layers and open water evaporation from the surface water layer. For non-paddy irrigation, the irrigation demand is calculated





using the difference between total and available water in the first two soil layers where total water is equal to the amount of water between field capacity and wilting point and available water is equal to the amount of water between current status and wilting point. Water withdrawal is calculated using the water efficiency rate of FAO (2012) and Frenken and Gillet (2012).

**Livestock water demand**

Livestock water demand is assumed to be the same as livestock water consumption and is calculated by the number of livestock in a grid cell with the daily drinking water requirement per individual livestock type (six livestock types in total) and per air temperature for seasonal change in drinking water requirement. The approach is taken from Wada et al. (2011,2014).

**Industrial and domestic water demand**

Calculation of industrial water demand also follows the method of Shen et al. (2008), Wada et al. (2011) using the gridded industrial water demand data for 2000 from Shiklomanov (1997) and multiplying it by water use intensity. Water use intensity is a function of gross domestic product (GDP), electricity production, energy consumption, household consumption, and a technological development rate per country. Domestic water demand is calculated by multiplying the population in a grid cell by a country-specific per capita domestic water withdrawal rate taken from FAO (2007) and Gleick et al. (2009). Adjustments for air temperature and for country-based economic and technological development are carried out based on the approach of Wada et al. (2011).

**Water withdrawal and return flows**

The approach to calculating water withdrawal from different sources, water consumption, and return flows is based on the work of de Graaf et al. (2014), Wada et al. (2014), Sutanudjaja et al. (2018) and Hanasaki et al. (2018). Water demand can be fulfilled by surface water and groundwater. Based on the work of Siebert et al. (2010) groundwater for irrigation can be only used in areas that are equipped for irrigation. Groundwater is, at first, only abstracted from the renewable groundwater storage. Water demand that cannot be fulfilled purely from groundwater uses surface water from rivers, reservoirs, and lakes. An environmental flow cap can be set in order to sustain environmental needs for rivers, reservoirs, and lakes. If water demand still cannot be fulfilled, additional water is taken from nonrenewable groundwater. At 5' resolution, water demand cannot always be covered by surface or groundwater resources in the same grid cell; therefore, CWatM uses the approach of LISFLOOD (Burek et al., 2013) and takes water from up to five grid cells downstream moving along the local drainage direction.

Return flow and associated losses (i.e., conveyance, application) are calculated using the approaches of LPJmL (Rost et al., 2008) and H08 (Hanasaki et al., 2018). Return flow is the flow which is withdrawn from surface water or groundwater but is not consumed. For the return flow rate we follow the approach of Hanasaki et al. (2018). For irrigation the return flow is calculated using the irrigation efficiency by Döll and Siebert (2002). For domestic and industry the return rate is based on Shiklomanov (2000) (i.e., 90% for the industrial sector and 85% for the domestic sector). Fifty percent of return water from



irrigation is lost to evaporation and 50% is returned to the channel network. This assumption is taken from Hanasaki et al. (2018). Domestic and industrial return flow is returned 100% to the river channel network.

## 3 Data

### 3.1 Mask map

CWatM can be run globally at 0.5° (30' or ≈ 50x50 km²) or 5' (≈ 10x10 km²) but also at a regional scale on 30', 5', or even on 30", as long as the mask map is specified. To speed up the runs, a set of coordinates or a mask map can be defined to run CWatM locally but using a global dataset. The use of netCDF format facilitates this operation.

### 3.2 Global datasets

Various global datasets were used to set the framing conditions for CWatM. The model provides full global datasets for the 30' and the 5' resolution. For both resolutions, sub-grid variability is considered for certain processes; for example, for snow the sub-grid variability of elevation is used, and for the effect of land cover, the sub-grid variability of land use/cover in each grid cell is used. Table 1 gives an overview of the global datasets. Further descriptions of these datasets are given in the supplement.

**Table 1: Global dataset, source of dataset and submodule of CWatM**

| Dataset | Source | Original spatial resolution | Submodule in CWatM |
|---|---|---|---|
| **Elevation** | SRTM (Jarvis et al., 2008); Hydro1k (USGS, 2002) | 3'', 1km | Snow |
| **Flow direction map** | DDM30 (Döll and Lehner, 2002); DRT (Wu et al., 2011) | 30'' | Routing, lakes |
| **Lakes and reservoirs** | HydroLakes database (Messager et al., 2016;Lehner et al., 2011) | Shapefile | Lakes, routing |
| **Soil** | Harmonized World Soil Database 1.2 (HWSD) (FAO et al., 2012) | 30'' | Soil |
| **Soil pedotransfer** | Rosetta3 (Zhang and Schaap, 2017) | - | Soil |
| **Groundwater** | GLHYMPS (Gleeson et al., 2011, 2014;Huscroft et al., 2018) | | Groundwater |
| **Land cover** | Forest land cover (Hansen et al., 2013) Imperious area (Elvidge et al., 2007) Irrigated areas (Döll et al., 2002b; Siebert et al., 2005, 2010) Hyde 3.2 database (Klein Goldewijk et al., 2017) | 1'' 30'' 5' 5' | Soil, land cover, water demand |
| **Crop coefficient** | MIRCA2000 (Portmann et al., 2010) | 5' | Soil, water demand |
| **Albedo** | GlobAlbedo dataset (Muller et al., 2012) | 3' | Pot. evaporation |
| **Discharge** | GRDC (Global Runoff Data Centre, 2007) | Station | Calibration |



| Population and GDP | Hyde 3.2 database (Klein Goldewijk et al., 2017) SSP Database at IIASA (Riahi et al., 2017) SSP population and GDP projections: Spatial disaggregation on 30' and 5' (Jones and O'Neill, 2016; Gao, 2017, Kummu et al., 2018 and Gidden et al., 2018) | 5' Country 7.5', 30'' | Water demand |
|---|---|---|---|
| Livestock water demand | Gridded livestock densities (FAO, 2007, Steinfeld et al., 2006) Livestock per country (FAO, 2012) | 5' | Water demand |
| Industry water demand | Gridded industrial water data (Shiklomanov, 1997) | 5' | Water demand |
| Domestic water demand | domestic water withdrawal per capita (FAO, 2012; Gleick et al., 2009) | 5' | Water demand |
| Meteorological forcing | WFDEI.GPCC (Weedon et al., 2014) PGMFD v.2 - Princeton (Sheffield et al., 2006) GSWP3 (Kim et al., 2012) MSWEP (Beck et al., 2017) EWEMBI (Lange, 2018) For downscaling to 5' WorldClim version2 (Fick and Hijmans, 2017) | 30' 30' 30' 6' 30' 30'' | Almost all |

## 4 Calibration

Most of the global hydrological models are uncalibrated with few exceptions, for example, WaterGAP (Müller Schmied et al., 2014). One of the main reasons for calibrating a model is the uncertainty of its input data, parameters, model assumptions, and grid cell heterogeneity, especially at low resolution as, for example, 0.5° or even 5'. Samaniego et al. (2017) gives a good overview of the main challenges to improving model parametrization. Calibrating CWatM is of major importance, as the model is developed to quantify water demand versus availability for detailed regional water resources assessments that will act as the

basis for interactions with stakeholders and regional policy development. For assessments of water resources and water demand and consumption such as these, realistic simulations of water resources use and availability are necessary.

The main challenge of global calibration is not only the large uncertainty of input data, and the lack of data and validation data, but also that the hotspots of water crisis occur in data-poor regions such as Africa and parts of Asia. For CWatM,

calibration uses an evolutionary computation framework in Python called DEAP (Fortin et al., 2012). DEAP implemented the evolutionary algorithm NSGA-II (Deb et al., 2002) which is used here as single objective optimization.

As objective function we used the modified version of the Kling-Gupta Efficiency (Kling et al., 2012), with r as the correlation coefficient between simulated and observed discharge (dimensionless), β as the bias ratio (dimensionless), and γ as the

variability ratio.





$$\text{KGE'} = 1 - \sqrt{(r-1)^2 + (\beta-1)^2 + (\gamma-1)^2} \tag{30}$$

where: $\beta = \dfrac{\mu_s}{\mu_o}$ and $\gamma = \dfrac{CV_s}{CV_o} = \dfrac{\sigma_s/\mu_s}{\sigma_o/\mu_o}$

where CV is the coefficient of variation, $\mu$ is the mean streamflow [$m^3\ s^{-1}$], and $\sigma$ is the standard deviation of the streamflow [$m^3\ s^{-1}$]. KGE', r, $\beta$, and $\gamma$ have their optimum at unity. The KGE' measures the Euclidean distance from the ideal point (unity)

of the Pareto front and is therefore able to provide an optimal solution which is simultaneously good for bias, flow variability, and correlation. For a discussion of the KGE objective function and its advantages over the often used Nash–Sutcliffe Efficiency (NSE) or the related mean squared error, see Gupta et al. (2009) and Hrachowitz et al. (2013).

The calibration uses a general population size ($\mu$) of 256, a recombination pool size ($\lambda$) of 32. The number of generations is set to 30, which we found was sufficient to achieve convergence for stations. The calibration parameters are listed in Table 2.


**Table 2: Calibration parameters (with flexibility to adjust the number and different parameters)**

---

Snow:

    1.   Snowmelt coefficient in [m/C°/day] as a degree-day factor

Evapotranspiration:

    2.   Crop factor as an adjustment to crop evapotranspiration

Soil:

    3.   Soil depth factor: a factor for the overall soil depth of soil layer 1 and 2

    4.   Preferential bypass flow: empirical shape parameter of the preferential flow relation

    5.   Infiltration capacity parameter: empirical shape parameter b of the ARNO model

Groundwater:

    6.   Interflow factor: factor to adjust the amount which percolates from interflow to groundwater

    7.   Recession coefficient factor: factor to adjust the base flow recession constant

        (the contribution from groundwater to baseflow)

Routing:

    8.   Runoff concentration factor: a factor for the concentration time of run-off in each grid-cell

    9.   Channel Manning's n factor: a factor roughness in channel routing

Reservoir & lakes:

    10.  Normal storage limit: the fraction of storage capacity used as normal storage limit

    11.  Lake A factor: factor to channel width and weir coefficient as a part of the Poleni's weir equation

    12.  Lake and river evaporation factor: factor to adjust open water evaporation

---





# 5 Results

## 5.1 Computational Performance of CWatM

With a daily time step, a global run of 100 years takes around 12 hours. That is 7.2 minutes per year (on a Linux single node - 2400 MHz with Intel Xeon CPU E5-2699A). For the global setting, soil processes are the most time-consuming part, taking 50% of all computing time, followed by routing with 25% and runoff concentration with 10%.

**Table 3: Computational time for a 0.5° global run in sequence of hydrological process (rain to river) and module setup**

|   | Process | ∑ % runtime 0.5° version |
|---|---|---|
| 1 | Read meteo. data | 6.2 |
| 2 | Evaporation pot. | 7.6 |
| 3 | Snow | 8.8 |
| 4 | Soil | 59.4 |
| 5 | Groundwater | 59.5 |
| 6 | Runoff concentration | 70.1 |
| 7 | Lakes | 70.4 |
| 8 | Routing | 95.5 |
| 9 | Output | 100.0 |


A basin run, for example, for the Rhine basin which is 160,800 km$^2$ in size, using a mask map from the global dataset (netCDF map sets) needs 40 minutes (0.5°) and 3 hours (5') for 100 years. That is 24 seconds per year for the 0.5° and 110 seconds per year for the 5' versions. For the Rhine basin, reading input maps takes up 79%, which is by far the most time-consuming process, followed by routing (kinematic wave) 10% and soil processes (8%).

**Table 4: computational time for a 0.5° and 5' run – Rhine basin (same as Table 3)**

|   | Process | ∑ % runtime 0.5° version | ∑ % runtime 5' version |
|---|---|---|---|
| 1 | Read meteo. data | 79.4 | 86.4 |
| 2 | Evaporation pot. | 80.5 | 87.5 |
| 3 | Snow | 80.9 | 87.9 |
| 4 | Soil | 88.8 | 89.8 |
| 5 | Groundwater | 88.9 | 92.9 |
| 6 | Runoff concentration | 89.6 | 93.6 |
| 7 | Lakes | 89.8 | 94.8 |
| 8 | Routing | 99.6 | 99.6 |
| 9 | Output | 100.0 | 100.0 |





## 5.2 Global water balance

The main global water balance components are calculated for the period 1979–2016 with the standard deviation of interannual variation. The spatial extent is from 90° N to 60° S. The Global 0.5° run uses a non-calibrated global standard parameter set.
The meteorological forcing uses the WFDEI data (Weedon et al., 2014). Table 5 shows the estimated global water balance components. Global average annual precipitation is around 125,000 km³/yr, which is 850 mm per year (assuming the CRU land mask and the WGS84 ellipsoid). Average runoff is 51,000 km³/year and average actual evaporation is 71,700 km³/yr. This is in the range of other global hydrological models (Haddeland et al., 2011). The runoff fraction is 0.42, which is at the lower end compared to other models (Haddeland et al., 2011), but can be explained because CWatM takes into account
evaporation from lakes and rivers. Groundwater recharge amounts to 19,000 km³/yr, which is higher than some of the GHMs (Mohan et al., 2018), such as WaterGAP or FAO statistics, but lower than PCR-GLOBWB2 (Sutanudjaja et al., 2018) or MATSIRO (Koirala et al., 2012). Figure 2 shows the spatial distribution of discharge and groundwater recharge which is similar to the distributions shown in Koirala et al. (2012) and Mohan et al. (2018).

**Table 5: Global water balance components over the period 1981-2016 simulated by CWatM**

| | Variable | Estimate [km³/yr] ± 1δ | Compared to other studies [km³/yr] |
|---|---|---|---|
| **Water balance** | Precipitation | 125,100 ±3000 | |
| | Runoff | 51,800 ±1880 | 42393 [1] range: 42,000-66,000 [4] |
| | Evaporation | 71,700 ±1880 | 65754 [1] range: 60,000-85,000 [4] |
| | Δ water storage | 1,600 ± 760 | |
| **Groundwater** | Ground water recharge | 19,000 ± 920 | 27,756 [1]  13,466 [2] range: 12,666-29,900 [3] |
| **Withdrawal by sector** | Agricultural | 2,000 range: 1250 - 2400 | 2735 [1] |
| | Domestic | 430 range: 270 - 590 | 380 [1] |
| | Industrial | 900 range: 680 – 1130 | 798 [1] |
| | Total | 3,330 range: 2200 - 4200 | 3,912 [1] |
| | Return flow | 950 range: 750 – 1150 | 1546 [2] |
| **Withdrawal by source** | Surface water | 2,650 range: 2060 – 3100 | 3172 [1] |
| | Groundwater | 680 range: 610 - 950 | 737 [1] range: 570-952 [2] |

[1] Sutanudjaja et al., 2018
[2] Hanasaki et al., 2018
[3] Mohan et al., 2018
[4] Haddeland et al., 2011

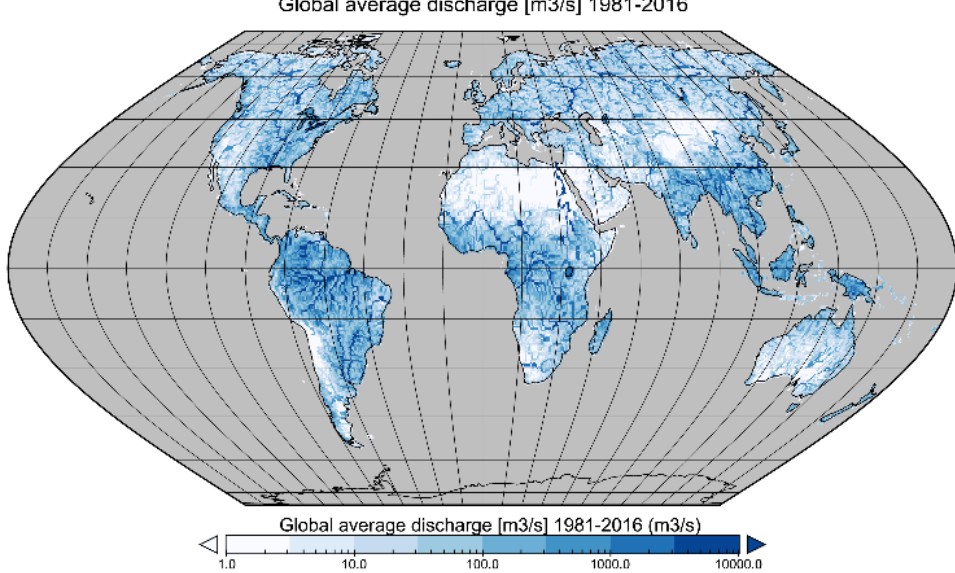

**Figure 2: Average global discharge in m³/s (1979-2016)**

It is important to note that water withdrawals from the agricultural sector (irrigation and livestock), industry, and domestic

sector (households) has been increasing over the years. The range in Table 5 for domestic and industry withdrawals has been

rising constantly from 1981 to 2016. Agricultural withdrawals have been increasing over time but achieved their maxima

during globally warm years, for example, 2002, 2009, and 2012. Water withdrawal from either surface water and groundwater

is within the range of other models. It has also been affected by the increasing water withdrawal for agriculture, industry, and

households.

**5.3 Global calibration results**

For calibration an evolutionary algorithm with KGE as objective function was applied and WFDEI meteorological data were

used as forcing. For all stations, the calibration improved the streamflow simulations compared to the baseline simulation with

a default parameter set. However, the performance varied depending on the quality of the discharge data and the meteorological

forcing, as well as on the processes included in CWatM, as shown in Table 6. Simulating processes such as backflow or large

evaporation losses due to swamps in the Nile and Niger basin are still challenging. But this simulation shows the suitability of

CWatM for representing the major water balance components and the necessity of calibrating certain basins, especially where

water availability is being compared with water withdrawal. A further step in global calibration must be performed by

regionalization of model parameters, for example, by using model parameters from well-performing basins for basins with

similar climate and other characteristics (Samaniego et al., 2010, 2017, Beck et al., 2016). A big challenge is the unevenly

distributed observed discharge data around the world with big spatial gaps in Africa and Asia. Even if calibration with an

objective function based on observed discharge is the best option, the gap might be filled with some sort of Budyko calibration





(Greve et al., 2016), where at least the empirical function of actual evapotranspiration against potential evaporation is fitted or satellite-based river levels could replace discharge missing from the observations (Revilla-Romero et al., 2015, Gleason et al., 2018).

**Table 6: Calibration results for some catchments worldwide**

| Continent | Catchment | Station | Calibration period* | Result for 30' | Results for 5' |
|---|---|---|---|---|---|
| **Europe** | Rhine | Lobith Germany Area: 160,800 km² | 1995-2010 [1] Uncal KGE: 0.55 (30') Uncal KGE: 0.58 (5') | KGE: 0.91 NS:   0.84 R²:   0.94 | KGE: 0.90 NS:   0.80 R²:   0.91 |
|  | Danube | Kienstock Austria Area: 95,970 km² | 1995-2010 [2] Uncal. KGE: 0.50 | KGE: 0.81 NS:   0.65 R²:   0.82 |  |
|  | Danube | Zimnicea Romania Area: 658,400 km² | 1995-2010 [1] Uncal. KGE: 0.61 | KGE: 0.84 NS:   0.64 R²:   0.87 |  |
| **America** | Yukon | Pilot station USA Area: 831,400 km² | 2001-2014 [3] Uncal. KGE: 0.54 | KGE: 0.63 NS:   0.50 R²:   0.83 |  |
|  | Sacramento River | Wilkins Slough USA Area: 33.500 km² | 1991-2010 [3] Uncal. KGE: 0.29 | KGE: 0.85 NS:   0.69 R²:   0.87 |  |
|  | Amazonas | Obidos Brasilia Area: 4,680,000 km² | 1985-1998 [1] Uncal. KGE: 0.43 | KGE: 0.89 NS:   0.80 R²:   0.91 |  |
| **Australia** | Murray River | Wakool Junction Australia Area: 78,000 km² | 2000-2012 [4] Uncal. KGE: -2.23 | KGE: 0.70 NS:   0.32 R²:   0.74 |  |
| **Africa** | White Nile | Jinja Uganda Area: 263,000 km² | 1996-2006 [5] Uncal. KGE: 0.43 |  | KGE: 0.94 NS:   0.90 R²:   0.95 |
|  | Zambezi | Lukulu Zambia Area: 206.500 km² | 1979-1989 [1] Uncal. KGE: 0.12 |  | KGE: 0.87 NS:   0.79 R²:   0.89 |
|  | Zambezi | Matundo-Cais Mozambique Area: 940,000 km² | 1979-1989 [1] Uncal. KGE: 0.33 |  | KGE: 0.57 NS:   0.14 R²:   0.57 |
| **Asia** | Olenek | 7.5KM Mouth of Pur Russia Area: 198,000 km² | 1991-1999 Uncal. KGE: 0.52 | KGE: 0.75 NS:   0.73 R²:   0.86 |  |
|  | Yangtze | Datong China Area: 1,705,400 km² | 2003-2013 Uncal. KGE: 0.54 | KGE: 0.84 NS:   0.69 R²:   0.87 | KGE: 0.90 NS:   0.75 R²:   0.90 |

Data for calibrating discharge from:
[1] GRDC, Global Runoff Data Centre, https://www.bafg.de/GRDC
[2] viadonau, viadonau Österreichische Wasserstrassen-Gesellschaft, http://www.viadonau.org
[3] USGS, United States Geological Survey, https://www.usgs.gov
[4] MDBA, Murray–Darling Basin Authority, https://riverdata.mdba.gov.au
[5] Ministry for Water and Environment, Uganda, https://www.mwe.go.ug



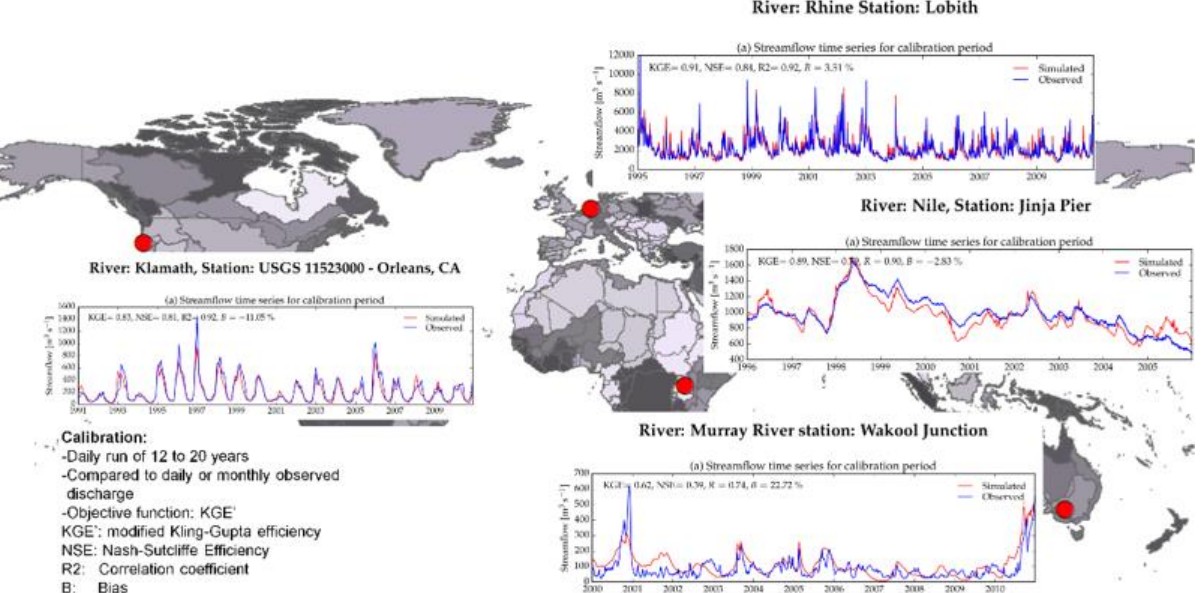

**Figure 3: Calibration results for some chosen stations globally**

## 5.4 Regional water balance: Example of East Africa

### 5.4.1 The extended Lake Victoria basin

The essential component of the Water Futures and Solution Initiative of IIASA (Burek et al., 2016, Wada et al., 2016) is the assessment of the balance water supply and demand for the present and into the future. With he support of the Government of Austria through the Austrian Development Agency (ADA) we aim to provide a deeper understanding of critical parameters for achieving water security in East Africa in the context of competing demands for basic water supply and sanitation, food security, economic development, and the environment. UN-Water (2013) p. 1 defines water security as:

> *"The capacity of a population to safeguard sustainable access to adequate quantities of and acceptable quality water for sustaining livelihoods, human well-being, and socio-economic development, for ensuring protection against water-borne pollution and water-related disasters, and for preserving ecosystems in a climate of peace and political stability."*

Water security is also a key ambition expressed in the "Vision 2050" of the East African Community (EAC, 2016) as rapid growth of the economy and population, and a high rate of urbanization are expected for the region and will lead to increased water demand in all sectors as well as further pressure on the water quality status.

The examples of operational areas for CWatM in this paper are not presented with specific results in mind, nor do they reflect results from the project's intensive stakeholder processes. They are there to demonstrate the value of a global hydrological model used in a regional case study that combines the temporal and spatial scale dependencies of water systems produced



through a scenario analysis designed to include both the regional and global scales. An "East Africa Regional Vision Scenario (EA-RVS)" was developed (Tramberend et al., 2019), based on regional visions, and we used available regional scenarios and

data that were developed in the context of global studies. As well as regional visions, the study also integrates into the widely applied global scenario development process of the Intergovernmental Panel on Climate Change (IPCC). It is characterized by a Scenario Matrix Architecture (van Vuuren et al., 2014) including the community-developed Shared Socioeconomic Pathways (SSPs) (Jiang and O'Neill, 2017) and the Representative Concentration Pathways (RCPs) (van Vuuren et al., 2011) for the characterization of climate change.

The study area, the extended Lake Victoria Basin (eLVB), is a transboundary basin in the tropics. It comprises the headwaters of the Nile and includes an area of over 460,000 km$^2$. The equator crosses the region approximately in the middle of the eLVB just south of Kampala. The eLVB includes the source of the Nile, major lakes in East Africa, foremost Lake Victoria, Lake Albert, Lake Edward, and Kyoga Lake. The eLVB has been subdivided into interconnected sub-basins. According to the water flow regime, we have aggregated the 61 basins into 8 major basins (see Figure 4). The CWatM model setup uses the default

global dataset on 5 arc min. Discharge data for calibrating river discharge were made available courtesy of the Ministry of Water and Environment, Uganda.

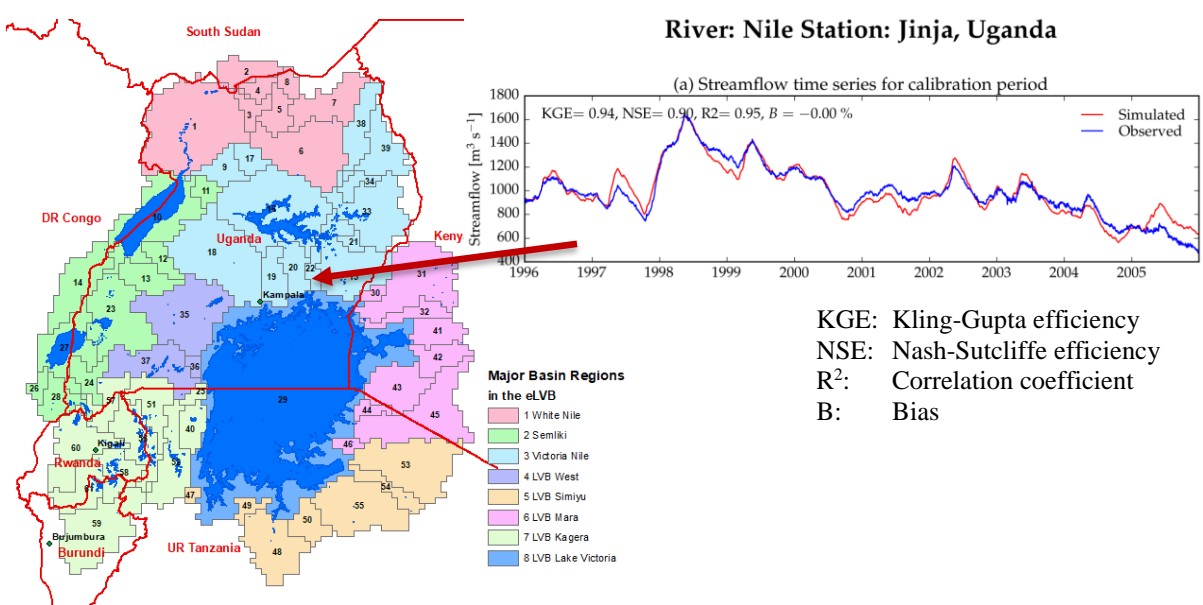

**Figure 4: The 61 sub-basins of the eLVB and their aggregation to 8 major basin regions**

**5.4.2 Seasonal pattern of the discharge regime**

For assessing climate change impact, RCP 6.0 was chosen as the most plausible future for East Africa by the "EAC Vision 2050" (EAC, 2016) even though it represents a rather pessimistic outlook of global temperature increases despite being published after the Paris Climate Agreement of 2015. We have chosen the two General Circulation Models (GCMs) of




HadGEM2-ES and MIROC5 out of the four GCMs (see Table 7) used in ISIMIP 2b (Frieler et al., 2016) as being the most feasible for eLVB as the discharge results that were run with CWatM for the historical runs of the GCMs GFDL-ESM2M and
IPSL-CM5A-LR showed a large discrepancy from historical results.

**Table 7: General Circulation Models (GCM)**

| GCM | Resolution | Institute | Nation |
|---|---|---|---|
| HadGem2-ES | 192 x 145 | Met Office Hadley Centre | UK |
| IPSL-CM5A-LR | 96 x 96° | Institut Pierre-Simon Laplace | France |
| GFDL-ESM2M | 144 x 90 | NOAA Geophysical Fluid Dynamics Laboratory | United States |
| MIROC-ESM-CHEM | Gaussian 128 x 64 | JAMSTEC, AORI, University of Tokyo, NIES | Japan |

Discharge is the variable which incorporates all the meteorological and hydrological processes in a basin and encompasses all
the storage components in a basin (i.e., soil, groundwater, lakes and reservoirs, etc.) Especially with the large lakes in the basin, discharge in eLVB has a long memory of past conditions.

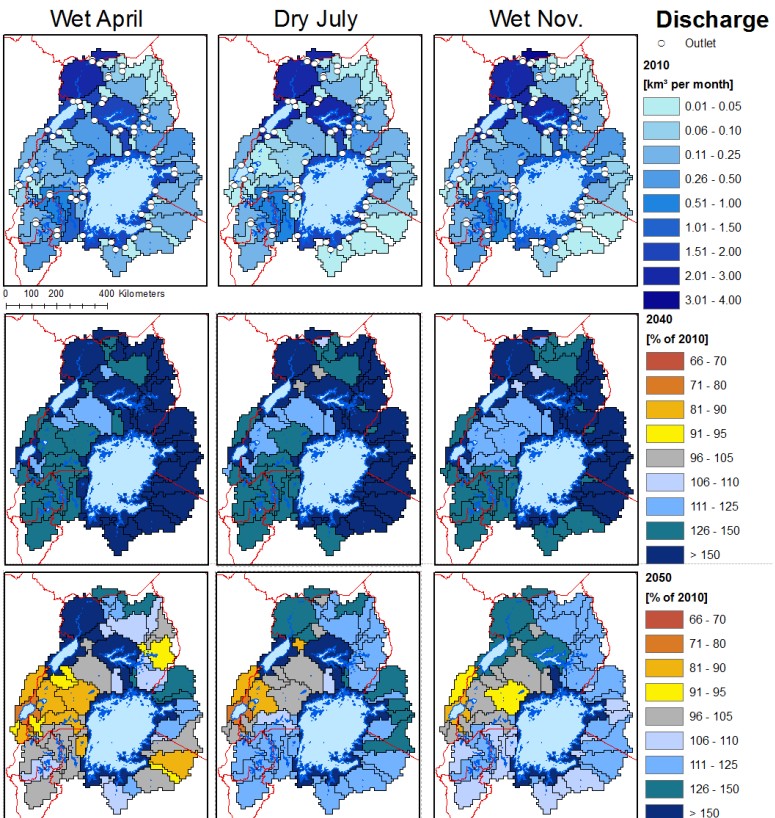

**Figure 5: Change of seasonal discharge pattern from 2010 to 2040 and 2050**





The seasonal pattern of discharge in Figure 5 shows more discharge for 2040 (10 year period 2036-2045) and 2050 (10 year period 2046-2055) in the river system from Lake Victoria, especially for the 2040 period. This is due to a wetter period of weather in the two GCMs from 2038 to 2049 and the strong memory effect of groundwater and the lakes. It also shows the big influence of inter-annual variability in the eLVB. Even if a general trend of less runoff in the 2050 period can be detected, long lasting periods of wetter conditions can nevertheless be superimposed over this trend. Because of the strong inter-annual

variability in the lower latitudes, it is difficult to assess the effect of a general climate change impact towards a wetter or drier climate. But under climate change, southwestern Uganda will show generally drier conditions than the western part of the eLVB.

### 5.4.3 Water scarcity indicators

Available water resources per capita, the Water Crowding Index (WCI), also called the Falkenmark indicator, is one of the

most widely used measures of water stress, (Falkenmark et al., 1989). Based on per capita water availability, the water conditions in an area can be categorized into different categories of stress expressed as m³ of water available per capita and year. Another indicator is the Water Resources Vulnerability Index (Raskin et al., 1997) also known as Water Exploitation Index (WEI) (EEA, 2005), defined as the ratio of total annual withdrawals for human use to total available renewable surface water resources. Regions are considered as water-scarce if annual withdrawals exceed the percentage of annual supply (Alcamo

et al., 2003). The thresholds for both indicators are shown in Table 8.

**Table 8: Water Crowding Index and Water Exploitation Index**

| Category | Water Crowding Index [m³ per capita per year] | Water Exploitation Index Water withdrawal / water availability [%] |
|---|---|---|
| no stress | > 1700 | < 20 |
| Stress | > 1000 - 1700 | ≥ 20 |
| Scarcity | 500 - 1000 | ≥ 20 |
| absolute scarcity | ≤ 500 | ≥ 40% |

The WCI and WEI are mainly shown as annual indicators, but in regions with high intra-annual variability, the rainy seasons

show a different picture from that of the dry season. An example, Figure 6, shows the WCI and WEI for the dry season and the most water-scarce month, July, for 61 sub-basins of the extended Lake Victoria basin, comparing the situations of 2010 and 2050. The figure shows that there is a clear increase in the WCI index. While in the current situation (2010), about half of the sub-basins are exposed to some level of water scarcity with some sub-basins indicating absolute water scarcity, in 2050 almost all sub-basins that are neither directly crossed by the River Nile nor adjacent to a lake, experience stress or scarcity,

many of them absolute water stress. The water resource availability for the WEI index is also based on RCP 6.0 climate scenario and includes the effect of human consumption and effects of land use change up to 2050. Looking at this index for the month of July only, it shows that nine out of 61 sub-basins are likely to experience water scarcity and even severely water-





scarce situations by 2050. Such sub-basins are mainly located at the south and southeastern shores of Lake Victoria and in densely populated areas of Rwanda and Burundi.


Interestingly, the WEI shows a much lower signal of water scarcity compared to the WCI. The WCI assumes that, regardless of the socioeconomic conditions, every person on the globe has the same "water demand entitlement." The Water Exploitation Index is based on the in situ situation and on balancing changing water availability and water demand. The fact that both indices show a rather different picture might be interpreted as an indication of economic water scarcity. The situation of low

economic development for the extended Lake Victoria basin may still prevail in 2050 (at least compared to the global average). This is the main reason for the relatively low actual water demand compared to global averages and therefore relatively low water scarcity signal for the WEI compared to the WCI.

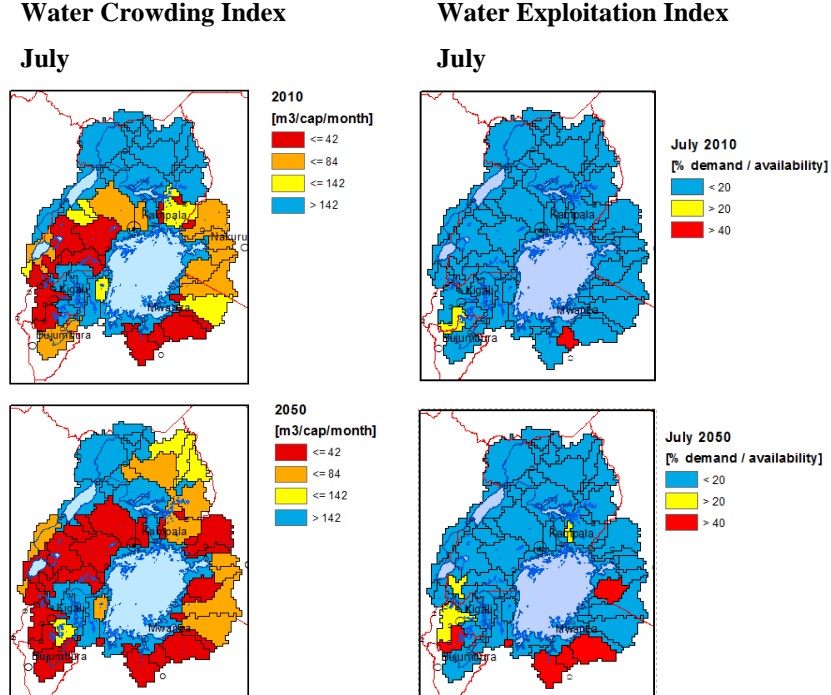

**Figure 6: Water Crowding Index and Water Exploitation Index in July for extended Lake Victoria Basin**

**5.5 Regional water balance: Example of the Zambezi**

**5.5.1 Calibration and comparison with other GHMs**

The hydrological model CWatM is intended to be scalable and can be applied over finer spatial scales (e.g., the basin). CWatM has been calibrated for the Zambezi, using six sub-catchments and measured discharge provided by the Global Runoff Data





Centre (2007). Figure 7 shows two time series of measured vs. simulated river discharge, and the comparison shows good
agreement of the modeled discharge with the measured data.

By comparing the outputs of the hydrological model ensemble, we see that, especially for sub-Saharan Africa, there is a strong
overestimation of river discharge, which indicates an erroneous picture if compared, for example, to water demands for
calculating water scarcity. Figure 8 shows a comparison of discharge for the Lukulu in the Zambezi basin of different
hydrological models.
The GHMs in Figure 8 use the WFDEI data (Weedon et al., 2014) as forcing meteorological data from 1981 to 2004. Apart
from WaterGAP and CWatM (both calibrated) one can see a strong overestimation of discharge for all other models compared
to the observed discharge.

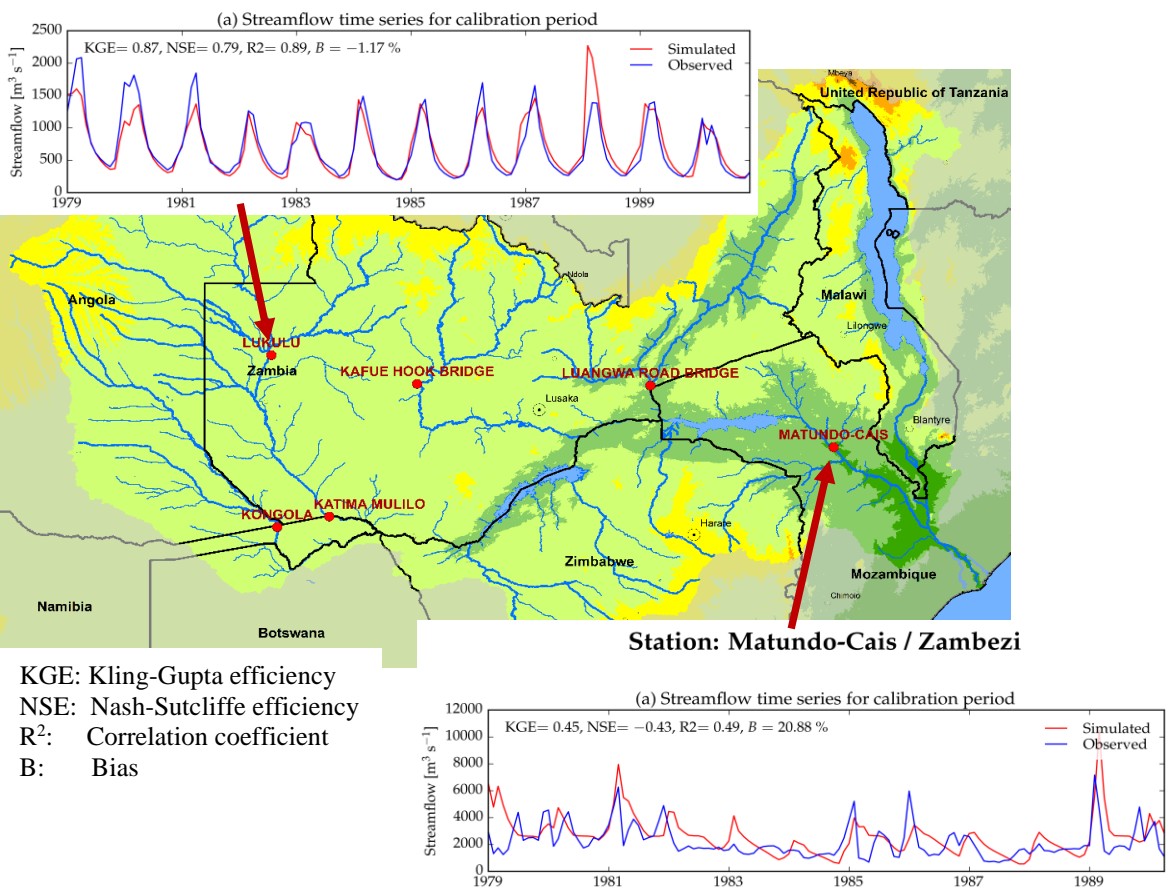

**Figure 7: Calibration results for two stations in the Zambezi basin**

Average discharge is overestimated for the non-calibrated models from two up to three times and maximum discharge up to
seven times. This shows the need to put efforts into calibration of the hydrological model for regional applications to be in line





with measured water resources and to minimize the uncertainty from hydrological modeling. Setting up model calibration has been time-consuming but inevitable for the Zambezi case study.

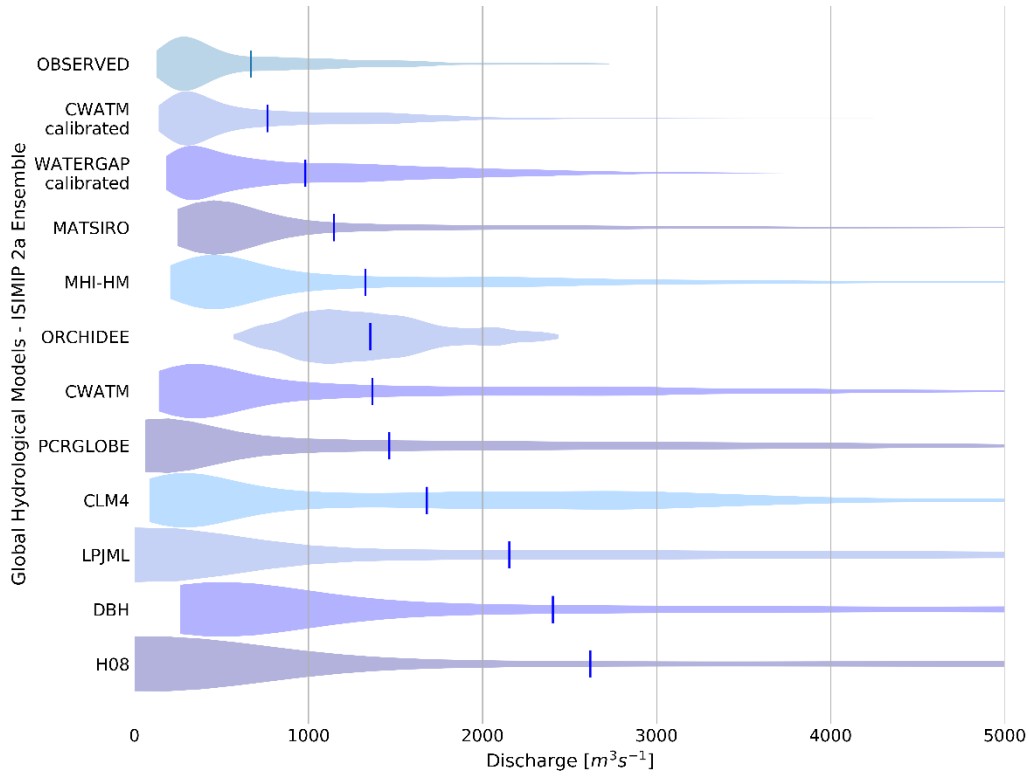

**Figure 8: Discharge for Lukulu/Zambezi from 1981–2004 for 11 different global hydrological models from the ISIMIP 2a ensemble compared with observed discharge. Each violin shows the probability density at different values of a GHM. The lines show the average discharge for each model**

### 5.5.2 Assessment of water stress

In a second phase, the CWatM calibrated model is used to assess water scarcity until 2050 in the Zambezi basin. Water resources at each grid cell are dependent on climate, water management (e.g., reservoirs) and water use for irrigation, livestock, domestic, or industry.

For each cell (at 5 arc min) (see Figure 9) and for aggregated regions, water resources can be related to water demand from different sectors. Results from the distributed hydrological model CWatM are aggregated to 21 sub-basins (see Figure 10) based on regional distribution shared by the Zambezi Water Commission (http://zamwis.wris.info). In addition, the regions of Kariba, Kafue, and Tete are split into, respectively, four, two, and four sub-basins, to look specifically into the more densely populated areas




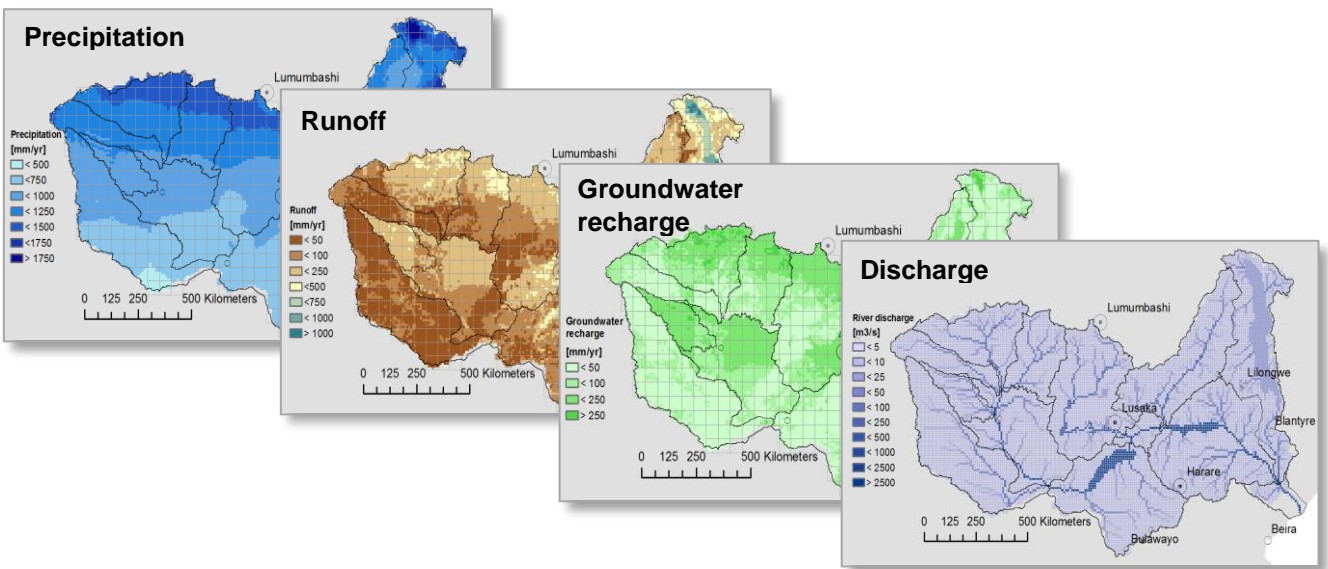

**Figure 9: Parameter sets of different hydrological variables**

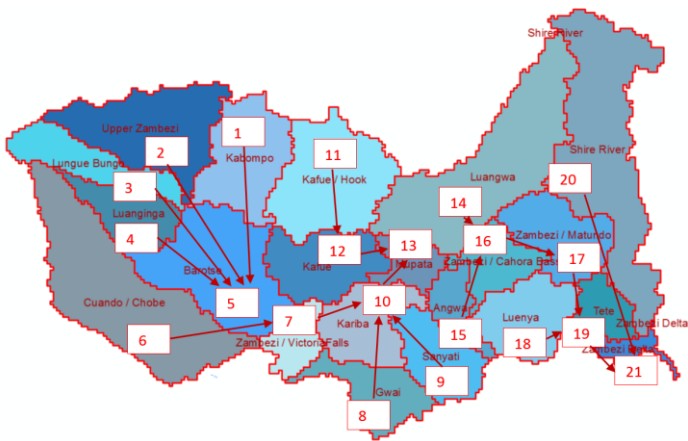

**Figure 10: Sub-basins of the Zambezi basin for aggregating data from CWatM**

Projection of future water resources builds on quantifications of climate scenarios CMIP5 (Distributed by the Coupled Model Intercomparison Project (CMIP), see http://cmip-pcmdi.llnl.gov/cmip5) based on the RCPs from the Inter-Sectoral Impact Model Intercomparison Project (ISI-MIP) (Frieler et al., 2016). We applied climate change projections from four GCMs (see Table 7) for a first setting of RCP6.0. Land use data projection is used from the GLOBIOM model (Havlík et al., 2013). Nineteen different crop types with different classes of farming intensity and eight land use classes (e.g., forest, build up classes)

of GLOBIOM output, for different RCPs and SSPs, are transformed to fit into the arrangement of six land use classes of CWatM.

Water demand for agriculture is taken from calculations within CWatM. Water demand for domestic, livestock, and industry
is calculated within CWatM using the approach of Wada et al. (2011,2014). The socioeconomic background needed for this
approach use data and methods for spatial disaggregation for the SSP2 scenario from Jones and O'Neill (2016), Gao (2017),
Klein Goldewijk et al. (2017), Kummu et al. (2018) and Gidden et al. (2018).

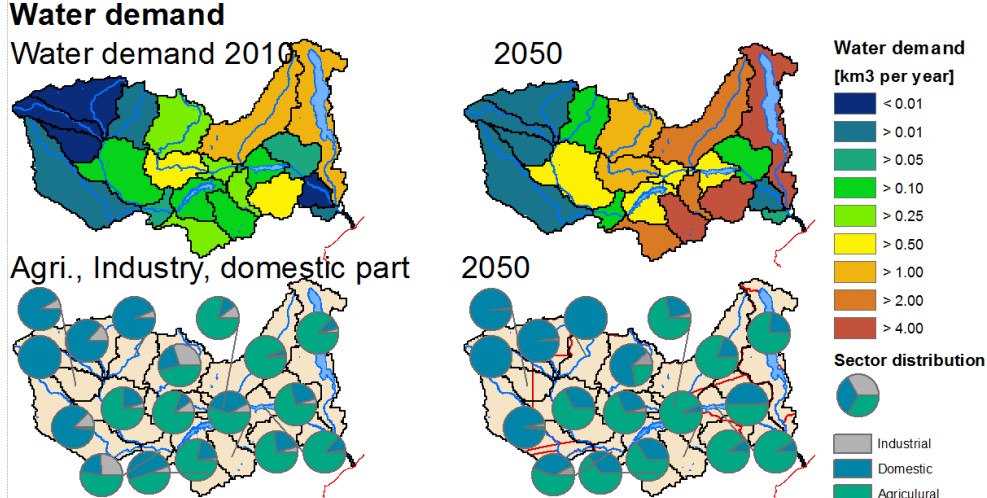

**Figure 11: Water demand projection for scenario SSP2/RCP6.0 to 2050 based on population, GDP, irrigation area projections**


**Water Exploitation Index for Zambezi**

The WEI is defined in Falkenmark et al. (1989), Falkenmark (1997) and Wada et al. (2011) as comparing blue water availability
with net total water demand. A region is considered "severely water stressed" if the WEI exceeds 40% (Alcamo et al., 2003).
The yearly WEI in figure12 shows no water stress for the whole basin in 2010 but water stress will intensify up to 2050 for
the business-as-usual (BAU) scenario (composed of the SSP2 and RCP6.0 scenarios), mainly due to agricultural and domestic
water demand increasing by a factor of five, as annual mean river discharge is only increasing by 6%. August is chosen for
monthly comparison as this is the month with the highest rate of water withdrawal (WW) and a mean monthly discharge
(MMD) that is only slightly higher than in November. The eastern part of the Zambezi basin, except for the main course of the
Zambezi River, was already showing severe water stress in 2010. This will increase in 2050 but the western part is still not
suffering water stress.





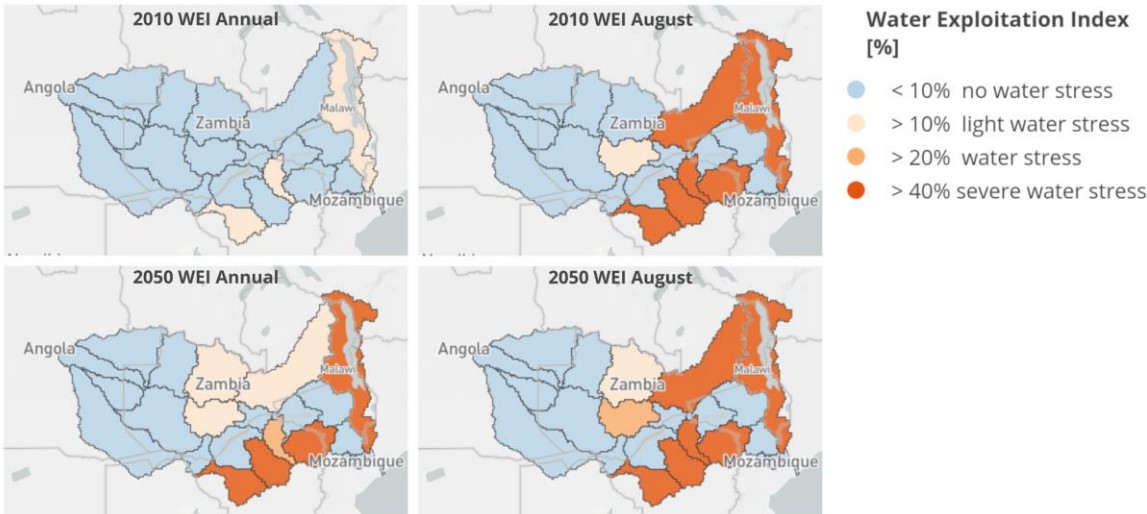

**Figure 12: Water Exploitation Index for 21 regions of the Zambezi, for 2010 and 2050 using the business-as-usual (BAU) scenario, yearly and for the month of August**

### 5.5.3 Coupling with hydro-economic and water quality models

As water stress and regional disparity is increasing in future, an integrated assessment framework is needed to explore potential sustainable pathways for the Zambezi basin. Therefore CWatM provides data on water availability (runoff and discharge) and
water demand (irrigation, domestic, and industrial demands) at sub-basin level to the "Extended Continental-scale Hydroeconomic Optimization" (ECHO) model (Kahil et al., 2018) and to the water quality model "Model to Assess River Inputs of Nutrients to seas" (MARINA) (Strokal et al., 2016). Figure 13 gives an overview of the interactions between models and the data flow.

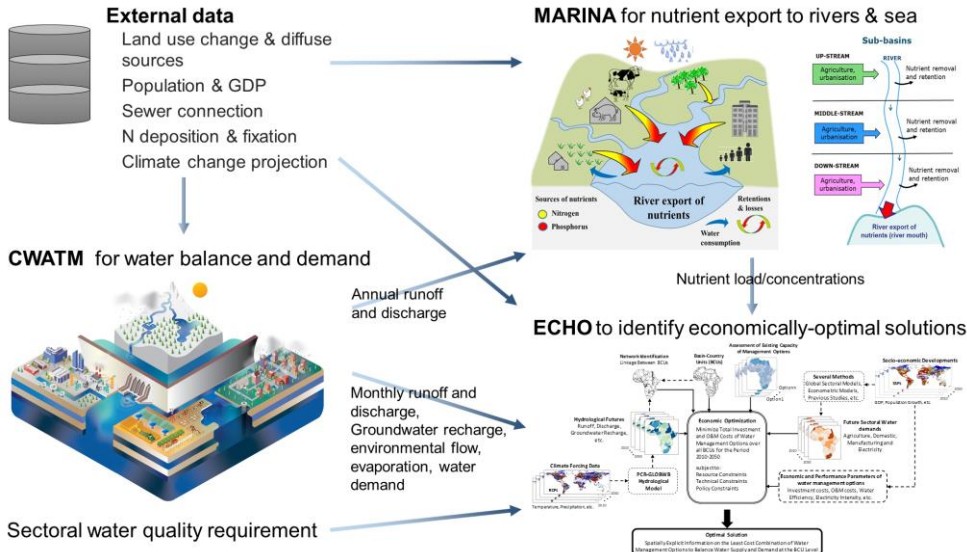

**Figure 13: Schematic view of the interaction among CWatM, Echo and MARINA**



ECHO is a hydro-economic optimization model. Its objective function minimizes the costs of water management options subject to several resource and management constraints across sub-basins within river basins over a long-term planning horizon (e.g., a decade or more). ECHO includes a wide range of supply- and demand-side water management options spanning over the water, energy, and agricultural systems. The supply options are surface water diversion, groundwater pumping, desalination, and wastewater recycling technologies. Other supply options considered in ECHO are surface water reservoirs and inter-basin transfer infrastructure. The water demand management options consist of different technologies for irrigation (flood, sprinkler, and drip), and several measures to improve crop water management in irrigation and water use efficiency in the domestic and industrial sectors (Kahil et al., 2018).

To assess the impacts of human activities on water quality, the MARINA model (Strokal et al., 2016) is used to estimate nitrogen loads and concentrations. MARINA quantifies nutrient (nitrogen and phosphorus) export to rivers and sea at the sub-basin scale. It is primarily used for long-term trend analysis and for source attribution, which could guide the identification of effective policy and management measures to reduce water pollution.

Moreover, MARINA uses data from GLOBIOM (Havlík et al., 2013) for land use and agricultural nitrogen inputs to the basin, and socioeconomic projections (population and GDP) to estimate nitrogen inputs from human waste. ECHO uses information on existing capacities of various water management options and the costs of investment in and operating these options. Nitrogen loads and concentrations calculated by MARINA are compared with nitrogen standards for different sectors to categorize the suitability of water use by different users, which can be further used by ECHO to optimize water allocation and explore economically optimal management options. The source attribution at the sub-basin scale by MARINA (Figure 13) provides prior information for ECHO to prioritize the most relevant nitrogen management options for each sub-basin, such as sewer connections, wastewater treatment, and manure and mineral fertilizer use in agriculture. Lastly, the coupling of MARINA and ECHO with CWatM enables analysis of the impacts of climate change and variability on nutrient export, water allocation, and adaptation costs. CWatM outputs from different climate forcing could be used in MARINA and ECHO to investigate the impacts of intra-basin spatial variability and inter-annual temporal variability of runoff and discharge. Figure 14 is an example of MARINA output of total dissolved nitrogen (TDN) in kg km$^{-2}$ yr$^{-1}$ for the Zambezi River basin. It illustrates the increase in river export of TDN to the sea between 2010 and 2050 (BAU scenario), the increasing share of anthropogenic nitrogen sources, and high spatial variability in the Zambezi basin (Tang et al., 2019). Another example of data exchange between CWatM and MARINA is given in Wang et al. (2019) for Lake Taihu in the Yangtzekiang basin.





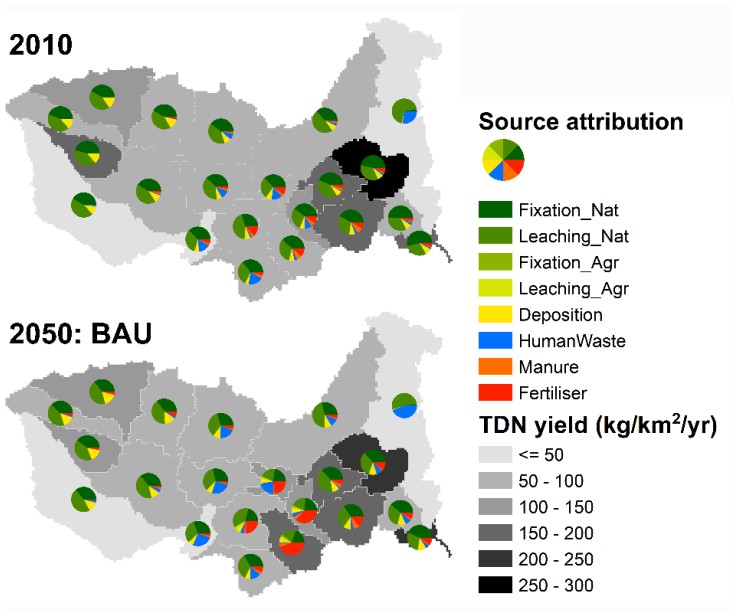

**Figure 14: Increase in river export of total dissolved nitrogen to sea between 2010 and 2050 (business-as-usual scenario)**

Figure 15 is an example of ECHO simulation results. It shows the costs for water supply and management in order to satisfy sectoral water demands (irrigation, livestock, domestic, and industrial) and environmental constraints (i.e., minimum environmental flow requirements and groundwater sustainability constraints) in the Zambezi river basin over the 2010–2050 period.

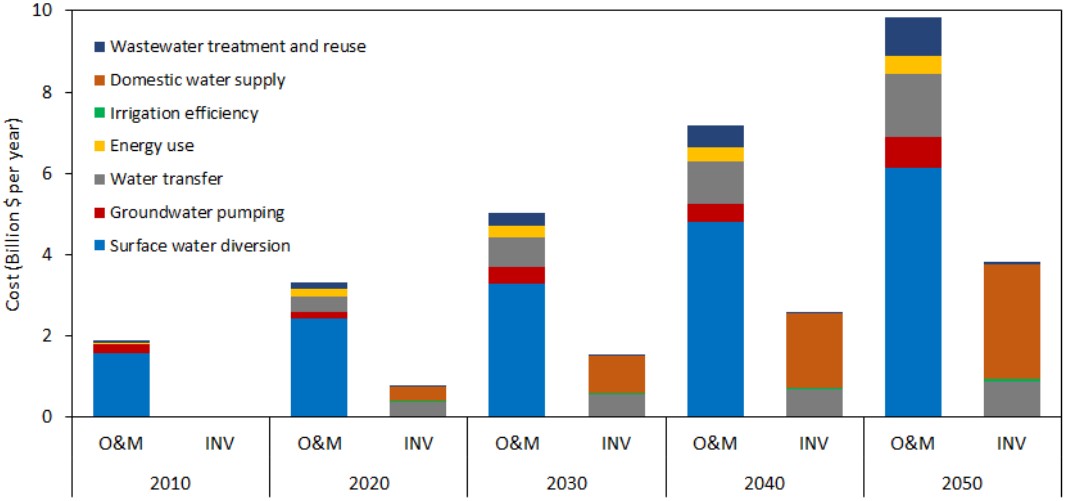

**Figure 15: Investment (INV) and operating (O&M) costs for water supply and management in the Zambezi basin between 2010 and 2050 (business-as-usual scenario)**





## 6 Conclusion and future work

We presented the new global hydrological model CWatM, which can be used globally and regionally at different resolutions with different datasets. The model is open-source in the Python environment and has a flexible modular structure. It uses
global, freely available data in the state-of-the art format of netCDF4 files to store and produce data in a compacted way. It includes major hydrological processes but also takes human water use into account by calculating water demand, water consumption, and return flows. Reservoirs and lakes are included in the model scheme. CWatM is being developed to include a routing scheme related to reservoirs and canals to better simulate water availability in both agricultural and urban contexts. It is shown that CWatM can be used in the framework of ISIMIP as a global model and also as part of a model integration of
hydrological, hydro-economic, and water quality models for assessing and evaluating water management options. This study also presented the need for a hydrological model to be calibrated to be able to estimate a detailed regional balance of water demand and water availability.

An external limitation and a source of uncertainty is the quality of meteorological forcing driving the hydrological models. As
shown in Müller Schmied et al. (2014) there are still discrepancies among the CMIP5 datasets and among the datasets and observations. Using CMIP6 datasets, (Eyring et al., 2016) is expected to reduce uncertainty. Another external model limitation and source of uncertainty is the availability of gauging station data, which is generally globally decreasing, completely unavailable , or difficult to access for some parts of the world. Continuous, consistent, and long-term river discharge data as an integral parameter over the whole basin are essential for basin modeling, water resources management, and flood
forecasting. Although the model represents the key hydrological processes, the groundwater model is relatively simple. But groundwater assessments (e.g., Bierkens et al., 2019) are becoming more and more important, as also is the importance of including lateral processes that increase the resolution of the model. Some other hydrological processes representation , for example, evaporation from swamps, namely, the Sudd in the Nile basin and the Niger river swamps need to be improved. The main direction of improvement should be better representation of human activities, for example, management of reservoirs,
including intra- and interbasin water transfer, and improving water demand requirements from agricultural sector by including irrigation schemes and plant phenology.

Future work will include 1) intensifying the development of a full dynamic coupling with a 2-D groundwater model, 2) developing a global calibration scheme that also takes sparse observation of discharge into account, 3) a finer resolution setting
for 1 km working for the upper Bhima basin in India as part of the Food–Water–Energy for Urban Sustainable Environments project (https://fuse.stanford.edu) supported by the Belmont Forum, and 4) an interdisciplinary project aimed at better understanding the effect of certain nexus policy interventions and solution options linked to ECHO and beyond.



## 7 Code and data availability

CWatM is written in Python 3.7 and C++ as an open source project under the term of the GNU General Public License
version 3. License and download information are on https://cwatm.iiasa.ac.at/license.html. The code can be used on different
platforms (Unix, Linux, Window, Mac) and is provided through a GitHub repository https://github.com/cwatm/cwatm. It
comes with the code, an executable program for Windows, a test case (River Rhine basin) and a settings file, and some tools
such as the calibration routine. The version of the model used to produce the results in this paper are stored as version 1.04 in
the GitHub repository and at Zenodo with the associated DOI https://doi.org/10.5281/zenodo.3361478 (Burek et al., 2019). A
global dataset on 0.5° and a dataset for the River Rhine are stored on https://doi.org/10.5281/zenodo.3361560.
Online documentation including documentation on the source code can be found on https://cwatm.iiasa.ac.at. Development
and maintenance of the official version of CWatM is conducted by the IIASA Water Program. Contribution, ideas, and users
are very welcome. Global data for 0.5° or 5' can be requested and stored on an IIASA ftp server.

**Acknowledgments**

The authors acknowledge the Global Environment Facility (GEF) for funding the development of this research and the CWatM
model development as a part of the Integrated Solutions for Water, Energy, and Land (ISWEL) project (GEF Contract
Agreement: 6993) and the support of the United Nations Industrial Development Organization (UNIDO). The authors also
acknowledge the continuous support of the Asian Development Bank (ADB), the Austrian Development Agency (ADA), and
the Austrian Federal Ministry of Sustainability and Tourism to the Water Futures and Solutions (WFaS) initiative at Water
Program of IIASA. This study and the model development were also conducted as part of the Belmont Forum Sustainable
Urbanisation Global Initiative (SUGI)/Food–Water–Energy Nexus theme for which coordination was supported by the US
National Science Foundation under grant ICER/EAR-1829999 to Stanford University. We appreciate all the other open source
projects which we used to collect ideas and which, on the other side, we hope to cross-fertilize with our ideas. We are very
grateful to all the freely available data sets. Any opinions, findings, and conclusions or recommendations expressed in this
material do not necessarily reflect the views of the funding organizations. This study is also partly supported by financial
support from the Austrian Research Promotion Agency (FFG) under the FUSE project funded by the Belmont Forum (Grant
Agreement: 730254), EUCP (European Climate Prediction System) project funded by the European Union under Horizon
2020 (Grant Agreement: 776613), and CO-MICC project which is part of ERA4CS, an ERA-NET initiated by JPI Climate
with co-funding by the European Union and the Austrian Federal Ministry of Science, Research and Economy (BMWFW).






**Supplement material**

1. Description of input data
2. Calibration results


**Authors contribution**

| | |
|---|---|
| Peter Burek | Writing original draft, software development |
| Yusuke Satoh | Software development (water demand), data curation |
| Taher Kahil | Methodology, writing (results part: linking to hydro-economic modeling), visualization |
| Ting Tang | Methodology, writing (results part: linking to water quality), visualization |
| Peter Greve | Software development (evaporation), visualization |
| Mikhail Smilovic | Software development (groundwater, water demand), visualization |
| Luca Guillaumot | Software development (groundwater MODFLOW coupling) |
| Yoshihide Wada | Conceptualization, funding acquisition, supervision, methodology,  reviewing writing |

is to the left of Ting Tang row.


**Competing interest**

The authors declare that they have no conflict of interest






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
