# Peer review of "Development of the Community Water Model (CWatM v1.04) A high-resolution hydrological model for global and regional assessment of integrated water resources management"

_Geoscientific Model Development, 2019_

## Referee Comment (RC1) · Anonymous Referee #1 · 1 Oct 2019

Burek et al. present the Community Water Model for integrated water hydrologic modeling and water resources management from the global to the regional scale. The authors claim that the novelty lies in the flexible modular approach for community development and application, the incorporation of hydro-socioeconomic components, and the user-friendly application from the global to the regional scale at high spatial resolution. I have a number of concerns that are expressed below.

The manuscript is not in the scope of GMD. The model concept (section 2.1) is described on 1.5 pages, and does not provide any insight into the modular model development approach, which should be the focus in GMD. Thus, the novelty can not be assessed. The theoretical descriptions of the hydrologic processes do not contain any new theoretical developments. The data sets used by the model are open and standard. The applied calibration procedure is also standard.

The results section does not allow any conclusion about the utility of the model. The input timing for the meteorologic information for a regional model does not lend confidence in the software development approach. The global water balance is not sufficient to demonstrate the utility of the model. There is much more information available to convince the reader of the usefulness of the proposed model. For example, Scanlon et al. (2018) posed a challenge for global hydrologic models to simulate correctly the water storage trends globally. The calibration results show that the calibration works, which would be surprising if not (are the parameters calibrated at each pixel?). But what about validation? While section 5.4 presents perhaps meaningful results, they are not suitable for GMD.

In addition, there is much more to community model development than the term and providing the code in a git plus documentation. I encourage the developers to study principles of best practices for community scientific software development and think about a software productivity and sustainability plan.

Scanlon, B. R. et al. Global models underestimate large decadal declining and rising water storage trends relative to GRACE satellite data. PNAS 115, E1080–E1089 (2018).

---

## Referee Comment (RC2) · Anonymous Referee #2 · 19 Oct 2019

Overall the paper was well structured and easy to follow. The paper did a good job of explaining the main concepts of the model and the key equations used while pointing the interested reader to the more detailed model documentation for further analysis. With a model like this where the science is not being advanced in itself it is challenging to highlight the novelty of the work, however, the authors do a good job of showing the utility of such a framework which allows modular integration with other sectors and climate models. Combined with the ability to work on various spatial resolutions down to 30 arc sec makes this a very promising tool. The particular challenge of dealing with

shared storage water bodies (groundwater, reservoirs, and lakes) while dealing with variable spatial scales and time periods is also a very difficult challenge that has been addressed in the paper. The analysis examples are relevant and useful for current and future issues and overall I believe the model has a lot of potential and recommend the paper for publication.

The model was easy to install and the example run solved without issues using the .exe version. The full data set was available for download at https://zenodo.org/record/3361560#.XapscehKhPY but did not include soil data and so gave an error that the "/cwatmdata/landsurface/soil/percolationImp.nc' file could not be found with a trial run of the Rhine example using the global dataset. I would also recommend adding more details about the climate data (as given in the meteo_wfdei_rhine_README.txt file) on the main tutorial webpage.

———————————————————

---

## Author Comment (AC1) · 4 Nov 2019

Thank you for your review of our paper. We are glad that you find our model valuable and that you recommend the paper for publication.

In this comment we report the reviewer's comments under <> brackets, followed by our replies.

<The full data set was available for download at https://zenodo.org/record/3361560#.XapscehKhPY but did not include soil data>

Thank you for testing and pointing out that we missed some data. We want to emphasis that a open source model is only useful, if the data set is also available. We looked again at the data set and uploaded a new and tested data set on https://doi.org/10.5281/zenodo.3528098 together with a climate set for the Rhine basin for testing purpose. We also changed the link in the paper pointing to the new data set.

< I would also recommend adding more details about the climate data >

The uploaded data includes a climate forcing data set cut to the Rhine basin and the description of the database include the text in the meteo_wfdei_rhine_README.txt. Several global climate forcing data are available for download on the web and no conversion is necessary because CWatM can read netdcf format. We added a sentence to the paper: Climate forcing data can be found on the ISI-MIP server (Frieler et al., 2016) or any other climate forcing data stored as netcdf can be used.

---

## Author Comment (AC2) · 19 Nov 2019

Thank you for your review of our paper.

In this comment we report the reviewer's comments under <> , followed by our replies.
* * *
<The manuscript is not in the scope of GMD. The model concept (section 2.1) is de-scribed on 1.5 pages, and does not provide any insight into the modular model devel-

opment approach, which should be the focus in GMD. Thus, the novelty can not be assessed. >

We have now expanded upon our modular model approach in section 2.1, and have described in greater detail the modeling structure and why we think our manuscript is both relevant and worth publishing. The targeted audience of the model are hydrological modelers of varying levels of programming familiarity. Modelers with no experience in programing languages like Python can use the executable together with the settings file. Modelers with only limited experience in Python can use the platform-independent Python version with no need to adapt the code itself. Finally, modelers with programming capacity in Python can engage with the source code and adapt the model to their specific needs. The wide adoption of Python as a programming language and the open source approach will allow for a community of developers to engage with and further develop CWatM.

CWatM follows a modular development pathway in several ways which should help to simplify the use of the model for the different user groups. 1) The program itself is independent from the settings file, which includes all relevant information of data, parameters for each process and output options. This enable the user to run the model without any understanding in Python and the users has a lot of flexibility to choose input data and output options. 2) The hydrological processes are modules separate modules from those related to data handling (e.g., reading configuration, reading and writing routines, error handling) and has individual modules for each hydrological process group (from the calculation of potential evaporation to river routing). This enables the advanced user to concentrate on developing their own hydrological modules. 3) In addition, each module is identically composed of an initialization class and a class which is run during the time steps. 4) CWatM keeps track of the metadata on sub routine of the source code and of data. It uses netCDF4 format as in- and output to store temporal-spatial data efficiently and to use meteorological forcing data as they come without reformatting. NetCDF4 also has the advantage that metadata are directly attached to the data. 5) In order to reach its community, CWatM comes with an online documentation on https://cwatm.iiasa.ac.at/, which includes the basic steps to setup the model, the data structure, the license information and uses Sphinx for auto-documentation of source code.

A major goal of the model is to link it to other models (see section 5.5.3). A modular structure is easier to maintain and can support a better interaction with these models. We have now expanded upon section 5.5.3 into section 6. Now called "Linking and integration with other sectoral models", we describe how the modular structure of CWatM can support better integration with other models, such as ECHO, MARINA, MESSAGE, GLOBIOM. We hope that this more accurately shows that the model takes care of using best practice in research software as stated in Wilson et al., 2014 and Jiménez et al., 2017. CWatM is already used in several scientific assessments including Wang et al., 2019a, Wang et al., 2019b, He et al. 2019, Vinca et al., 2019, Kahil et al. 2019.
* * *
< The theoretical descriptions of the hydrologic processes do not contain any new theoretical development. The data sets used by the model are open and standard. The applied calibration procedure is also standard. >

The utility and relevancy of our model is in its possible integration with other models and sectors, the capacity to model at multiple spatial scales, and its capacity to foster a community of use and development. Several hydrological processes are relatively new including preferential flow pathways, the timing of runoff concentration, and the urban extent as an independent land cover class. The hydrological processes and combination of systems are modeled according to referenced literature. The data sets are necessarily open and standard to comply with open source practices. The calibration is standard, but also flexible and adaptive. The model is thus offered as a package both immediately useable and adaptable. At the moment only WaterGAP is calibrating their model. We see the need (shown in Fig 8) to properly calibrate the model and even go

for a globally calibrated model. We appreciate the comment and have expanded the discussion section 2.1.

————————————————————————————————————————

< The results section does not allow any conclusion about the utility of the model >

To highlight some aspects of section 5, we provide examples of where the model has been applied in conjunction with water quality and hydro-economic modeling, with case studies and results for both the extended Lake Victoria basin the Zambezi basin. Further we highlight that CWatM has been applied for different purposes (Wang et al., 2019a, Wang et al., 2019b, He et al. 2019, Vinca et al., 2019, Kahil et al. 2019). We appreciate the comment and have expanded upon the discussion by shifting section 5.5.3 into section 6 to better underline the linkage to other models and results.

————————————————————————————————————————

< The input timing for the meteorologic information for a regional model does not lend confidence in the software development approach>

We are currently using a daily timesteps with daily meteorological forcing data. The model is also using daily sub-time steps where required e.g. the percolation in the soil and the routing, as well as the lake and reservoirs module uses sub-steps. For soil this is already in: "Therefore, the soil moisture equation has to be solved iteratively on a sub-daily time step." For river routing we have added: As water can travel a distance greater than a cell size in one day, river routing and the lake and reservoir routines are performed on a sub-daily time step, based on the chosen spatial resolution. The spatial scale we are using at the moment is 5 arcmin to 30 arcmin. Even models dealing with flood forecast on European level with a spatial resolution of 5x5 km2 (Lisflood in the European Flood Awareness System https://www.efas.eu/en/overview) are using daily time steps. We are aware that if we using a spatial resolution of 1 km2 (like in our project in India https://fuse.stanford.edu/hydrology-and-water-resources) and dealing

with flooding or even flash-flood we have to go down in timely resolution as well, but at the moment we work on water availability, water demand with daily available climate projections. ─────────────────────────────────────────

< The global water balance is not sufficient to demonstrate the utility of the model, There is much more information available to convince the reader of the usefulness of the proposed model. For example, Scanlon et al. (2018) posed a challenge for global hydrologic models to simulate correctly the water storage trends globally.>

We are aware that several papers and models use either observed discharge stations or GRACE to evaluate the global results of their models (see below for a list). For sure we can produce maps which shows some agreement between simulation and measurement but: 1) exactly the paper you mention Scanlon et al. (2018) points out that the used models show a "poor agreement between models and Grace" and to show some results for a single model might be not a real prove. 2) we think that it is best practice to show the performance of a model is in the framework of an intercomparison project in comparison with other models. As our model is quite young (one reason to have this manuscript here) we are not part of the study in Scanlon et al. (2018) but we are already part of Inter-Sectoral Impact Model Intercomparison Project (ISIMIP), and upcoming publication will include results from CWatM e.g. from Yadu Pokhrel et al. on global terrestrial water storage. This includes also a fair and independent comparison of GRACE and model results from seven global hydrological models.

Further on we showed in fig 8 that especially in data sparse regions uncalibrated results from different models can be quite wrong (also CWatM) and calibration or fitting is necessary. This is shown only for a single station to illustrate that, but if necessary several other stations can be shown. This is a different view of evaluation than showing global maps of correlation or other indicators but shows also some shortcoming of global models (and of CwatM). We appreciate the comment and the reference to Scanlon et al. (2018), and we are working on furthers comparisons for future work. We have added this comment into the manuscript: Some modeling papers such as

Döll et al., 2014 and Sutanudjaja et al., 2018 use observed discharge stations or the Gravity Recovery and Climate Experiment (GRACE) Tapley et al., 2004 to evaluate the global results of their models. As CWatM has started to be part of the ISIMIP inter-comparison project, we evaluated that it is best practice to show the performance of a model in the framework of ISIMIP by comparing it to other models like in Zhang et al., 2017 or Scanlon et al., 2018. An upcoming paper by Pokhrel, 2019 on global terrestrial water storage will include a comparison of seven global terrestrial hydrology models (including CWatM) against GRACE data.

——————————————————————————————————————

<The calibration results show that the calibration works, which would be surprising if not (are the parameters calibrated at each pixel?)>

The idea of showing calibration here is to show that it works. Most scientific papers show a method and shows afterward to what extent it works. Parameter are calibrated on sub-basins (for Lake Victoria and Zambezi). We have added this line in 5.5 to account for that: Calibration is performed for three stations. The calibration parameters are valid for the sub-basin up to the gauging station. The upstream station is calibrated using the best fit of the downstream calibrated sub-basins. Calibration for the Zambezi basin is performed for six stations (Lukulu, Kongola, Katima, Kafue Hook, Luangwa Road Bridge, Tete - see figure 6). The calibration parameters are valid for the sub-basin up to the gauging station. The upstream station is calibrated using the best fit of the downstream calibrated sub-basins. The parameter set is valid for the sub-basin exclusive of the downstream sub-basins which have their own parameter set.

——————————————————————————————————————

Validation of calibration results are show in the supplement. We added a line to refer to the supplement for validation.

——————————————————————————————————————————

<While section 5.4 presents perhaps meaningful results, they are not suitable for GMD.> <In addition, there is much more to community model development than the term and providing the code in a git plus documentation. I encourage the developers to study principles of best practices for community scientific software development and think about a software productivity and sustainability plan>

In fact there is a lot more than providing code and documentation. We take this point and show in section 2.1 that we include some more ideas based on established examples such as the Community Land Model (Lawrence et al. 2018) or the mesoscale Hydrologic Model (Samaniego et al. 2019). Several aspects of our supports the user and development, incorporating good practices for scientific software development, such as code management, documentation with a tutorial, and automated documentation. We already have users in China, Europe, Japan, India, and other countries, so there exists the demand for a model like CWatM. Some parts are still under development and can be improved like our forum on https://cwatm.iiasa.ac.at/forum.html where users can help each other. The fact that we do not have a full set "software productivity and sustainability plan" is because we are at the point of establishing our model and one part of this is to publish our model at a journal which promote an open research culture like GMD. But we use this comment to extend section 2.1. For sure there is still a lot to do to reach a high standard like improving software management by building up an automated testing, easier installation via the Python Package Index and building containers and improving the communication with the users via an improved forum.

---

## Author Response (AR1)

**Development of the Community Water Model (CWatM v1.04) A high-resolution hydrological model for global and regional assessment of integrated water resources management**

Peter Burek[1], Yusuke Satoh[1,2], Taher Kahil[1], Ting Tang[1], Peter Greve[1], Mikhail Smilovic[1], Luca Guillaumot[3] and Yoshihide Wada[1,4]

**Reply to reviewer 1**

Thank you for your review of our paper.

In this comment we report the reviewer's comments under <>, followed by our replies.

<The manuscript is not in the scope of GMD. The model concept (section 2.1) is described on 1.5 pages, and does not provide any insight into the modular model development approach, which should be the focus in GMD. Thus, the novelty can not be assessed. >

We have now expanded upon our modular model approach in section 2.1, and have described in greater detail the modeling structure and why we think our manuscript is both relevant and worth publishing.

Changes in the manuscript:

The target audience of the model is hydrological modelers of varying levels of programming familiarity. Modelers with no experience in programming languages like Python can use simply the executable together with the settings file. Modelers with only limited experience in Python can use the platform-independent Python version with no need to adapt the source code itself. Finally, modelers with programming capacity in Python can engage with the source code and adapt the model to their specific needs. The wide adoption of Python as a programming language and the open source approach will allow for a community of developers to engage with and further develop CWatM. The code itself comes with a GNU General Public License and is hosted on GitHub (https://github.com/CWatM/CWatM), where every change is trackable and transparent. The source code is programmed in the modern programming language Python, with only certain computationally demanding parts written in in C++, such as river routing. Each subroutine is documented for its design and purpose, and 40% of the source code lines are documentation.

CWatM follows a modular development pathway in several ways which simplify the use of the model for the different user groups. Firstly, the program is independent from the settings file, which includes all information related to data, parameters for each process, and output options. This enable the user to run the model without any understanding of Python, while still providing flexibility of input and output options to the user. Secondly, the modules for hydrological processes and data handling (e.g., reading configuration, data read and write routines, error handling) are handled separately, and further the different hydrological processes (from calculation of potential evaporation to river routing) are each handled independently.

This enables the advanced user to concentrate on adapting specific processes or developing their own hydrological modules to extend the modular structure (see figure S11 in the supplement for the CWatM modular structure). Thirdly, each module is identically composed of an initialization class and a dynamic class operating through time; this structure is motivated by the PC-Raster framework (Karssenberg et al., 2010). Fourthly, CWatM keeps track of the metadata on sub routines of the source code and of the data. CWatM generally accepts netCDF, Geotiff, and PCRaster input maps and uses netCDF4 formats for outputs and to store temporal-spatial data efficiently. This also allows for meteorological forcing data to be used without the need for reformatting. NetCDF4 also has the advantage that the metadata are directly attached. Finally, to best support and reach its community, CWatM has a Google group and forum (https://groups.google.com/d/forum/cwatm), online documentation (https://cwatm.iiasa.ac.at) including model setup basics, data information, license information, and uses Sphinx (https://www.sphinx-doc.org) for the auto-documentation of source code.

The model is accessible and customizable to the needs of different users with varying levels of programming skill, allowing for research questions of varying spatial scales from global to local scales to be answered. This will support and enable different stakeholder groups and scientific communities beyond hydrology and of varying capacities to engage with a hydrological model and support their investigations (see section 6). We hope that that we have appropriately represented CWatM and its use of best practices in research software as stated in Wilson et al., 2014 and Jiménez et al., 2017. CWatM has already used in several scientific assessments, including Wang et al., 2019a, 2019b, He et al. 2019, Vinca et al., 2019, and Kahil et al., 2019, and has a small but growing community users in several countries around the world.

We changed section 5.6 to 6 and added this to the manuscript:

6 Linking and integration with other sectoral models

The modular structure of CWatM helps to link and integration with other models. The independent setting file offers possibilities to adapt the input and output to other models. For a lot of applications no intervention into the code is necessary. If code has to be customized to the linked model, the modular structure of CWatM eases to identify the point of intervention. To explore potential sustainable pathways for the Zambezi basin, an integrated assessment framework is needed.

< The theoretical descriptions of the hydrologic processes do not contain any new theoretical development. The data sets used by the model are open and standard. The applied calibration procedure is also standard. >

The utility and relevancy of our model is in its possible integration with other models and sectors, the capacity to model at multiple spatial scales, and its capacity to foster a community of use and development. Several hydrological processes are relatively new including preferential flow pathways, the timing of runoff concentration, and the urban extent as an independent land cover class. The hydrological processes and combination of systems are modeled according to referenced literature. The data sets are necessarily open and standard to comply with open source practices. The calibration is standard, but also flexible and adaptive. The model is thus offered as a package both immediately useable and adaptable. At the moment only WaterGAP

is calibrating their model. We see the need (shown in Fig 8) to properly calibrate the model and even go for a globally calibrated model. We appreciate the comment and have expanded the discussion section 2.1.

< The results section does not allow any conclusion about the utility of the model >

To highlight some aspects of section 5, we provide examples of where the model has been applied in conjunction with water quality and hydro-economic modeling, with case studies and results for both the extended Lake Victoria basin the Zambezi basin. Further we highlight that CWatM has been applied for different purposes (Wang et al., 2019a, Wang et al., 2019b, He et al. 2019, Vinca et al., 2019, Kahil et al. 2019). We appreciate the comment and have expanded upon the discussion by shifting section 5.5.3 into section 6 to better underline the linkage to other models and results.

Changes in the manuscript:

CWatM has already used in several scientific assessments, including Wang et al., 2019a, 2019b, He et al. 2019, Vinca et al., 2019, and Kahil et al., 2019

< The input timing for the meteorologic information for a regional model does not lend confidence in the software development approach>

We are currently using a daily timesteps with daily meteorological forcing data. The model is also using daily sub-time steps where required e.g. the percolation in the soil and the routing, as well as the lake and reservoirs module uses sub-steps.

For soil this is already in: "Therefore, the soil moisture equation has to be solved iteratively on a sub-daily time step."

For river routing we have added in the manuscript:

As water can travel a distance greater than a cell size in one day, river routing and the lake and reservoir routines are performed on a sub-daily time step, based on the chosen spatial resolution.

The spatial scale we are using at the moment is 5 arcmin to 30 arcmin. Even models dealing with flood forecast on European level with a spatial resolution of 5x5 $km^2$ (Lisflood in the European Flood Awareness System https://www.efas.eu/en/overview) are using daily time steps. We are aware that if we using a spatial resolution of 1 $km^2$ (like in our project in India https://fuse.stanford.edu/hydrology-and-water-resources) and dealing with flooding or even flash-flood we have to go down in timely resolution as well, but at the moment we work on water availability, water demand with daily available climate projections. Our model is running from 20 to 100+ sub-daily timestep depending on the context and application.

< The global water balance is not sufficient to demonstrate the utility of the model, There is much more information available to convince the reader of the usefulness of the proposed model. For example, Scanlon et al. (2018) posed a challenge for global hydrologic models to simulate correctly the water storage trends globally.>

We are aware that several papers and models use either observed discharge stations or GRACE to evaluate the global results of their models (see below for a list). For sure we can produce maps which shows some agreement between simulation and

measurement but: 1) exactly the paper you mention Scanlon et al. (2018) points out that the used models show a "poor agreement between models and Grace" and to show some results for a single model might be not a real prove. 2) we think that it is best practice to show the performance of a model is in the framework of an intercomparison project in comparison with other models. As our model is quite young (one reason to have this manuscript here) we are not part of the study in Scanlon et al. (2018) but we are already part of Inter-Sectoral Impact Model Intercomparison Project (ISIMIP), and upcoming publication will include results from CWatM e.g. from Yadu Pokhrel et al. on global terrestrial water storage. This includes also a fair and independent comparison of GRACE and model results from seven global hydrological models.

Further on we showed in fig 8 that especially in data sparse regions uncalibrated results from different models can be quite wrong (also CWatM) and calibration or fitting is necessary. This is shown only for a single station to illustrate that, but if necessary several other stations can be shown. This is a different view of evaluation than showing global maps of correlation or other indicators but shows also some shortcoming of global models (and of CwatM). We appreciate the comment and the reference to Scanlon et al. (2018), and we are working on furthers comparisons for future work. We have added this comment into the manuscript:

Some modeling papers such as Döll et al., 2014 and Sutanudjaja et al., 2018 use observed discharge stations or the Gravity Recovery and Climate Experiment (GRACE) Tapley et al., 2004 to evaluate the global results of their models. As CWatM has started to be part of the ISIMIP intercomparison project, we evaluated that it is best practice to show the performance of a model in the framework of ISIMIP by comparing it to other models like in Zhang et al., 2017 or Scanlon et al., 2018. An upcoming paper by Pokhrel, 2019 on global terrestrial water storage will include a comparison of seven global terrestrial hydrology models (including CWatM) against GRACE data.

The calibration results show that the calibration works, which would be surprising if not (are the parameters calibrated at each pixel?).

The idea of showing calibration here is to show that it works. Most scientific papers show a method and shows afterward to what extent it works. Parameter are calibrated on sub-basins (for Lake Victoria and Zambezi). We have added these lines in 5.4 to account for that:

Calibration is performed for three stations. The calibration parameters are valid for the sub-basin up to the gauging station. The upstream station is calibrated using the best fit of the downstream calibrated sub-basins.

Calibration for the Zambezi basin is performed for six stations (Lukulu, Kongola, Katima, Kafue Hook, Luangwa Road Bridge, Tete - see figure 6). The calibration parameters are valid for the sub-basin up to the gauging station. The upstream station is calibrated using the best fit of the downstream calibrated sub-basins. The parameter set is valid for the sub-basin exclusive of the downstream sub-basins which have their own parameter set.

But what about validation?

Validation of calibration results are show in the supplement. We added a line to refer to the supplement for validation.

We added this in section 5.3

Calibration and validation results are shown for each station in the supplement part 2

<While section 5.4 presents perhaps meaningful results, they are not suitable for GMD.>

<In addition, there is much more to community model development than the term and providing the code in a git plus documentation. I encourage the developers to study principles of best practices for community scientific software development and think about a software productivity and sustainability plan>

In fact there is a lot more than providing code and documentation. We take this point and show in section 2.1 that we include some more ideas based on established examples such as the Community Land Model (Lawrence et al. 2018) or the mesoscale Hydrologic Model (Samaniego et al. 2019). Several aspects of our supports the user and development, incorporating good practices for scientific software development, such as code management,  documentation with a tutorial, and automated documentation. We already have users in China, Europe, Japan, India, and other countries, so there exists the demand for a model like CWatM. Some parts are still under development and can be improved like our forum on https://cwatm.iiasa.ac.at/forum.html where users can help each other.

The fact that we do not have a full set "software productivity and sustainability plan" is because we are at the point of establishing our model and one part of this is to publish our model at a journal which promote an open research culture like GMD. But we use this comment to extend section 2.1.

For sure there is still a lot to do to reach a high standard like improving software management by building up an automated testing, easier installation via the Python Package Index and building containers and improving the communication with the users via  an improved forum.

We added this to section 7:

and 5) improving software management by building up an automated testing, easier installation via the Python Package Index and building containers and improving the communication with the users.

**Reply to reviewer 2**

Thank you for your review of our paper. We are glad that you find our model valuable and that you recommend the paper for publication.

In this comment we report the reviewer's comments under <> brackets, followed by our replies.

<The full data set was available for download at https://zenodo.org/record/3361560#.XapscehKhPY but did not include soil data>

Thank you for testing and pointing out that we missed some data. We want to emphasis that a open source model is only useful, if the data set is also avail-able. We looked again at the data set and uploaded a new and tested data set on https://doi.org/10.5281/zenodo.3528098 together with a climate set for the Rhine basin for testing purpose. We also changed the link in the paper pointing to the new data set.

< I would also recommend adding more details about the climate data >

The uploaded data includes a climate forcing data set cut to the Rhine basin and the description of the database include the text in the meteo_wfdei_rhine_README.txt. Several global climate forcing data are available for download on the web and no conversion is necessary because CWatM can read netcdf format.

We added a sentence to the paper in section 8:

[revised manuscript text omitted]

---

## Author Response (AR2)

**Topical Editor Decision: Reconsider after major revisions** (04 Feb 2020) by Wolfgang Kurtz
Comments to the Author:

*Dear authors,*

*thank you very much for the revision of your manuscript. I checked the changes with respect to the reviewer comments and also asked one additional referee to provide comments on the manuscript.*

*From my perspective, some of the critical points from reviewer 1 still remain. The new statements on modularity/ novelty are still rather broad and general. More technical details on the software engineering approach should be provided (see specific comments below). As it is a model description paper, this is the right place to provide (and focus on) such technical details of the model implementation. With respect to novelty, it would also be helpful to relate the concept and implementation of CWatM to already existing global hydrological models. In addition, the validation deserves more attention and should be extended and placed more prominent in the paper.*

*Therefore, I would like to ask you to revise your manuscript according to the items listed below and the comments of reviewer 3.*

*Kind regards,*
*Wolfgang Kurtz*

We sincerely thank the editor and the additional referee for their comments. We agree with the overall perception that the manuscript could benefit from further clarification of technical details, better representation of the model's novelty, and on validation. We have thus extended the introduction and methods sections, and a included a section on validation. Please find a point-by-point response to each comment below.

*Specific comments:*
*- It should be clarified how the modularity exactly manifests in the software architecture. For example, if someone wants to use an alternative representation of a certain process (e.g., water transport in the soil column, river routing), how does the architecture of CWatM support the implementation of such new code in terms of modularity? Is there an API or specific data structures that support modularity? The mentioned separation of input specifications and computation (line 129-133) as well as the separation of (hydrological) processes in different subroutines/ modules is also common for many other models in my opinion. How does CWatM differ in that respect?*

Alternative descriptions of processes (e.g. Hargreaves, Thornwaite, instead of Penman-Monteith for calculating potential evaporation) can be included in a module as different initial and dynamic classes, and the selection of the specific process representation can be included in the settings file. An advanced user can add another python class elaborating another method. A user without python knowledge can then use this new method by selecting it in the settings file.

*- It is mentioned that CWatM can be easily coupled to other models (particularly due to its modularity). More details would be advantageous here. How does the architecture of CWatM support such model coupling? Is there a dedicated API for model coupling and how do the established model couplings work in detail? Is this done by some standardized (API-related) library calls or via transfer of information by specific input/output files (customized to each model separately)?*

Linking to other models can be done by transfer via input and output files where every global variable of CWatM (examples include evapotranspiration, lake and reservoir storage, etc.) can be written as annual, monthly, or daily time series as text files for specific points or aggregated to basins, or as maps showing the value for each cell. Any variable can have a metainformation entry. This enables a simplified linking to other models (e.g. hydro-economic) which might need only e.g. monthly values of groundwater recharge per basin. Linking to models like the land use model GLOBIOM is done with pre- and postprocessing coupler functions, as most of these models needs aggregated data as ASCII files. Coupling to MODFLOW (McDonald and Harbaugh, 1988, Harbaugh, 2005) is done by using the FloPy Python package (Bakker et al., 2016). The user can switch on the MODFLOW coupling in the settings file and in addition the necessary data for the groundwater model (e.g. transmissivity maps) have to be provided. Coupling to models using C++ can be done by an in-memory coupling using the ctypes library, as this is already done to embed the kinematic wave routing routine.

*- It is emphasized that usage of netCDF as a standard format is an advantage of CWatM, e.g. because of the possibility of metadata annotation. How is that realized in the program? Does CWatM e.g. use CF conventions for model output (as other hydrological models like WRF-Hydro)? Which metadata are (or can be) annotated to the model output?*

CWatM uses Climate and Forecast (CF) Metadata convention 1.6. Metadata information (e.g. unit, long name, standard name, author, etc.) can be automatically included for every output/parameter NetCDF file by adding this information to a file metanetcdf.xml.

*- It is also mentioned that CWatM can use different sources of meteorological input data sets (due to usage of netCDF). How does CWatM support the usage of these different data sources? Does it automatically care e.g. for different names of input variables in the different data sets?*

*Does it support the user by regridding the meteorological forcings to the model grid (if yes, which interpolation/regridding schemes are available)?*

As long as the forcing data are using the CF convention (here 1.6) CWatM takes care of the different names of the input variables and re-scale the dataset on to a different area of interest such as a catchment or global scale depending on a mask map or predefined rectangular. The forcing data are automatically regridded to the model grid (e.g. 30'' or 5') using the delta change method (Moreno and Hasenauer, 2016, Mosier et al., 2018) based on high resolution monthly data from WorldClim version2 (Fick and Hijmans, 2017)

*- Section 5.2, lines 609-614: Model description papers in GMD should also focus on a sufficient validation of the model. Given the fact that comparisons with discharge and/or GRACE data are already available, some of these results should also be included and discussed in this model description paper.*

*- Section 5.3: lines 619-620: Validation results should be discussed in more detail in the main part of the paper itself.*

We have now put in an additional section 5.3  Global model validation, where we compare the key model result (i.e., river discharge) against observed data (i.e., GRDC stations).

As CWatM has started to be part of the ISIMIP intercomparison project (ISIMIP2b), we think it is best practice to show the performance of a model in the framework of ISIMIP by comparing it to other models like in Zhang et al., 2017 or Scanlon et al., 2018. An upcoming paper by Pokhrel et al., 2020 on global terrestrial water storage (under review in Nature Climate Change) will include a comparison of seven global terrestrial hydrology models (including CWatM) against GRACE data. An another upcoming paper by  Telteu et al. (under preparation by ISIMIP Global Water sector) is assessing different processes in different Global Hydrological Models (incl. CWatM)  by comparing the equations for different modules and processes. This is more towards the diagnostics of large-scale hydrological models in order to understand the structural uncertainty of the models.

*- Figures 4+7: Discharge is shown for the respective calibration period. It would be worthwhile to additionally show results for an independent validation period.*

We have provided validation results in table 7 and in the supplement. Related to the African stations in Uganda and on the Zambezi with limited observed data, we have used all the observed data for the calibration process.

We have added in 5.5.1:

The ten years of available observed data are used for the calibration period. Therefore, no other time period is available for a validation period. The data limitation is still prevalent in

Africa and we will try to include more data in our future study if available from regional stakeholders.

*- The incorporated calibration feature is interesting and it would also be helpful to the reader to get some more information on expected simulation times for the calibration exercise mentioned in the manuscript.*

Added in section 4:

For the example of the Rhine catchment on 5' a single simulation of 20 years (5 years as spin up time and 15 years for comparing to observed data) takes around 40 minutes. After an initial 256 simulations for the general population, another 960 simulation are run (30 generation · 32 pool size). Together, these 1216 simulations are run on 32 nodes in parallel sessions in around 25 hours.

*- Tables 3+4: It would be helpful to also provide non-cumulative runtime percentages.*

Added the non-cumulative runtime percentages in tables 3 and 4.

**Review #3**

*General comments*
*The paper describes a new macroscale hydrological model called CWatM.*
*Although the model does not include much new functions compared with the*
*most advanced similar models, but it does have several advantages due to its*
*well organized code and adoption of modern technology. The unique feature of*
*CWatM is that although global model, it can be easily applicable to specific*
*basins. It also provides some tools for parameter tuning. These functions are*
*seldom seen in earlier global models (exception is H08, Mateo et al. 2014;*
*Masood et al. 2015). The text is well written and organized. The Section 5.4 and*
*5.5 are interesting, but too short to understand the methods and uncertainties in*
*results.*

We sincerely thank the referee for her/his overall positive evaluation of the manuscript and the helpful suggestions. Please find a point-by-point response to each comment below.

*Specific comments*

*Line 323-324 "With a simplification, the 1D Richards equation, is entirely gravity-driven and the matrix potential gradient is zero. This implies a flow that is always in a downward direction at a rate equal to the conductivity of the soil": It sounds like that the authors assume the water transport in unsaturated zone is same as saturated one. Is this really the case?*

The Richards equation is the basis for describing the soil moisture profile under unsaturated conditions. Unsaturated flow is different to saturated flow and one of the important differences between unsaturated and saturated flow is the differences in hydraulic conductivity. The Richards equation takes this into account as the hydraulic conductivity is a function of water content.
In order to clarify the referee's point, we have rewritten this paragraph:
In order to apply an analytical and faster solution the model of (Mualem, 1976) and the Van Genuchten model equation are used as a simplification of the 1D Richard equation. This implies a flow that is always in a downward direction at a rate equal to the conductivity of the soil.

*Line 487 "Irrigation water demand": Does CWatM consider multiple cropping? If so, how was it implemented?*
Yes, CWatM implicitly considers multiple cropping based on MIRCA2000 dataset, that include multi-cropping linking to crop calendar over a year.

To briefly summarize crop/irrigation water demand in CWatM, the crop-calendar, growing season, and monthly spatial distribution of crops are derived

from MIRCA2000 (Portman et al. 2010) -- MIRCA2000 explicitly considers multiple cropping. The associated crop- and stage-specific crop coefficients are derived from the Global Crop Water Model (Siebert et al. 2010). The crops are then aggregated into paddy and non-paddy and the crop coefficients are similarly aggregated by weighing the area of each crop class. Then, the cell-specific crop coefficient as it changes in time is related to the crops growing in this cell, inclusive of multiple cropping considered in the MIRCA2000 dataset. This methodology has been taken from Wada et al. (2014) and (2016).

*Table 1: The original spatial resolution for DDM30 must be 30'.*
Yes, Referee is correct that the original spatial resolution of DDM30 is at 30' and DRT from Wu et al. at 5'.
Thanks for pointing this out. We have now corrected this.

*Table 5 "Withdrawal of agricultural sector 2000 km3/yr:" I think the estimate is quite lower than that is shown in other reports. I am wondering this is partly due to the exclusion of multiple cropping.*
Two previous studies with WaterGAP (Siebert and Döll, 2010) and PCR-GLOBWB (Wada et al.,2011a, b)  using the same crop calendars and crop classes from MIRCA2000 estimate irrigation water withdrawals at 2099 (with Penman-Monteith) and 2057 km$^3$ respectively. Our estimate is relatively similar to these and this is confirming as the same crop information is used. Referring to the overview of irrigation water demands from Wada et al. (2016), these estimates are on the lower end of estimates, but still within the range. Estimates range from 1530-4100 km$^3$ and are generally between 2000-3000 km$^3$. Some discrepancy may also arise due to a high uncertainty in irrigation efficiency. Multiple cropping is implicitly included in CWatM (and in the MIRCA2000 dataset).

Siebert, S., Burke, J., Faures, J. M., Frenken, K., Hoogeveen, J., Döll, P., and Portmann, F. T.: Groundwater use for irrigation – a global inventory, Hydrol. Earth Syst. Sci., 14, 1863–1880, doi:10.5194/hess-14-1863-2010, 2010.
Wada, Y., van Beek, L. P. H., and Bierkens, M. F. P.: Modelling global water stress of the recent past: on the relative importance of trends in water demand and climate variability, Hydrol. Earth Syst. Sci., 15, 3785–3808, doi:10.5194/hess-15-3785- 2011, 2011a.
Wada, Y., van Beek, L. P. H., Viviroli, D., Dürr, H. H., Weingartner, R., and Bierkens, M. F. P.: Global monthly water stress: II. Water demand and severity of water, Water Resour. Res., 47, W07518, doi:10.1029/2010WR009792, 2011b.

Wada, Y., et al. "Modeling global water use for the 21st century: Water Futures and Solutions (WFaS) initiative and its approaches." *Geoscientific Model Development* 9 (2016): 175-222.

*Line 615 "Global calibration results": During the calibration, were the models for human activities (e.g. water abstraction, reservoir operation, and others) enabled?*

Thanks for pointing at this topic. Yes, human influence i) land use change ii) reservoirs iii) water abstraction were enabled. We have now mention this in the text.

*Line 640 "Section 5.4": This section is very interesting, but too short to understand the methods and uncertainties in results. For instance, I hardly understood how the lake, the dam (the Owen Falls Dam), and water use in the study domain were treated. Are there any reports which include more detailed information?*

Thank you for this positive comment. As much as we would like to include more details, the text length is quite limited in this type of model description paper for each case study. Thus, we would like to refer to our new study ofTramberend et al. (2020)  and a paper (under review) on this topic. This new study describes the full details of the model coupling, scenario development, human-natural components, and key model results using CWatM and ECHO models. The pre-print of Tramberend et al. (2020) is available publicly in the link below.

Tramberend, Sylvia and Burtscher, Robert and Burek, Peter and Kahil, Taher and Fischer, Günther and Mochizuki, Junko and Kimwaga, Richard and Nyenje, Philip and Ondiek, Risper and Nakawuka, Prossie and Hyandye, Canute and Sibomana, Claver and Luoga, Hilda Pius and Matano, Ali Said and Langan, Simon and Wada, Yoshihide, East African Community Water Vision. Regional Scenarios for Human - Natural Water System Transformations. ONE-EARTH-D-20-00017. Available at SSRN: https://ssrn.com/abstract=3526896 or http://dx.doi.org/10.2139/ssrn.3526896

*Line 640 Section 5.5: Same as above. Are there any reports which include more detailed information?*

Section 5.5 describes the regional applications of the model. The project reports include futher details of the model setup and the coupling to other sectoral models. These reports are available at  http://www.iswel.org/results/tools/ . We are also working towards peer-reviewed publication for this.

*Figure 7 the hydrograph for Matundo-Cais: It seems that the station is below the*

*massive reservoir of Cahora Bassa. The observation in Fig 7 apparently shows the influence of reservoir operation (i.e. the seasonal fluctuation during low flow period is "unnaturally" stable). The simulated hydrograph also shows "unnatural" behavior which is quite different from the observation. Here comes my two questions. First, did the author consider the reservoir operation of Cahora Bassa in this study? If this is the case, how was the reproducibility of the dam operation (the dynamics of storage and release)?*

1.)
Yes the reservoir operations of Cahora Bassa (and of Kariba dam) are included but the model tries to reproduce the reservoir operation in a general way (see 2.3.11). The real reservoir operation of Kariba dam and Cahora Bassa dam are not reproduced and therefore the model is good at reproducing the magnitude of discharge but not exactly the time series.

2.)
The producibility of the dam operation is done with some "simple" assumptions described in section 2.3.11.

3.)
We have added:
The station Matundo-Cais is downstream of the two big reservoirs Kariba Dam and Cahora Bassa which are included in the model. The reservoir operations are calculated with the approach of section 2.3.11

*Figure 8 I couldn't understand how to understand the "violin diagram". Elaborate what it shows.*
*Figure 8 Caption "at different values of a GHM": Unclear. What does it mean?*

The violin plot is similar to a box plot but they show the probability density of the data. While a box plot shows some statistics like mean and quartiles a violin box shows the full distribution of the data. Here it is used instead of a box plot to show the similarity between observed and different models. In general all uncalibrated model overestimate discharge for that station but apart from this some models also have a different shape than the observed data

[revised manuscript text omitted]

---

## Author Response (AR3)

**Topical Editor Decision: Publish subject to minor revisions (review by editor)** (06 May 2020) by
Wolfgang Kurtz

Comments to the Author:
Dear authors,

thank you very much for the revision of your manuscript which basically addressed the points from the previous review round.
After going through the paper again, I found a few minor issues that should be handled before the paper can be published:

- line 346-348: The statement here is not quite clear to me. The Mualem-Van Genuchten model provides closed-form equations for unsaturated hydraulic conductivity (equation 12) and the soil water retention curve. But their usage in equation 11 does not necessarily lead to the 'simplification' mentioned in the second sentence. Do you refer to the lower boundary condition (i.e. free drainage) in the lowest soil layer here? Please provide more details on how soil water flow is calculated in the model.

- line 601: Please clarify what a 'node' is in this context. Same as in line 608?

- Please check for consistent font size (e.g. lines 653-659), text color (line 738) and correct punctuation (e.g. line 263/264/877).

Kind regards,
Wolfgang Kurtz

We sincerely thank the editor for his comments:

- We replace line 346-348 with the following sentences and we hope that clarifies the soil water model.
  In order to apply an analytical and faster solution Van Genuchten (1980) hydraulic functions based on Mualem's (1976) model were adopted. It assumes a matric potential gradient of zero, which implies a flow that is that is always in a downward direction at a rate equal to the conductivity of the soil, and free drainage as the lower boundary condition in the lowest soil layer. The relationship between hydraulic conductivity and soil moisture status is described by the Van Genuchten (1980) equation.

- We replace 'node' with CPU core, which is the exacter term for this.
- Font size and text color were corrected for the lines and checked through the text
- Punctuation was set correctly
- Some other spelling, white spaces errors where corrected

Best regards,

Peter Burek